# A single N6-methyladenosine site regulates lncRNA HOTAIR function in breast cancer cells

Allison M. Porman[1], Justin T. Roberts[1,2], Emily D. Duncan[2,3], Madeline L. Chrupcala[1,4], Ariel A. Levine[1,4], Michelle A. Kennedy[1], Michelle M. Williams[5], Jennifer K. Richer[5], Aaron M. Johnson[1,2,4]*

1 University of Colorado Anschutz Medical Campus, Biochemistry and Molecular Genetics Department, Aurora, Colorado, United States of America, 2 University of Colorado Anschutz Medical Campus, Molecular Biology Graduate Program, Aurora, Colorado, United States of America, 3 University of Colorado Anschutz Medical Campus, Cell and Developmental Biology Department, Aurora, Colorado, United States of America, 4 University of Colorado Anschutz Medical Campus, RNA Bioscience Initiative, Aurora, Colorado, United States of America, 5 Department of Pathology, University of Colorado Anschutz Medical Campus, Aurora, Colorado, United States of America

* Aaron.m.johnson@CUAnschutz.edu

**Data Availability Statement:** All gene expression data associated with this publication are currently publicly available through GEO accession number GSE173530.

## Abstract

N6-methyladenosine (m6A) modification of RNA regulates normal and cancer biology, but knowledge of its function on long noncoding RNAs (lncRNAs) remains limited. Here, we reveal that m6A regulates the breast cancer-associated human lncRNA HOTAIR. Mapping m6A in breast cancer cell lines, we identify multiple m6A sites on HOTAIR, with 1 single consistently methylated site (A783) that is critical for HOTAIR-driven proliferation and invasion of triple-negative breast cancer (TNBC) cells. Methylated A783 interacts with the m6A "reader" YTHDC1, promoting chromatin association of HOTAIR, proliferation and invasion of TNBC cells, and gene repression. A783U mutant HOTAIR induces a unique antitumor gene expression profile and displays loss-of-function and antimorph behaviors by impairing and, in some cases, causing opposite gene expression changes induced by wild-type (WT) HOTAIR. Our work demonstrates how modification of 1 base in an lncRNA can elicit a distinct gene regulation mechanism and drive cancer-associated phenotypes.

## Introduction

Long noncoding RNAs (lncRNAs) are becoming increasingly noted for their roles in transcriptional regulation [1]. Members of this class of noncoding RNAs are typically longer than 200 nucleotides, transcribed by RNA polymerase II, and processed similarly to mRNAs [2]. LncRNAs regulate transcription in a variety of ways; they can direct histone-modifying enzymes to their target loci to induce changes in chromatin or can regulate transcription directly by interacting with transcription factors and RNA polymerase II [1]. Importantly, lncRNAs are often key regulators of epigenetic changes that can drive cancer progression, frequently via their aberrant overexpression [3,4].

The human lncRNA HOTAIR is a 2.2 kb spliced and polyadenylated RNA transcribed from the HoxC locus. Originally identified as a developmental regulator acting in *trans* to

**Funding:** This work was supported by the National Institute of General Medical Sciences (https://www.nigms.nih.gov) (R35GM119575 and R35GM144358 to A.M.J.; T32GM008730 to J.T.R.); the National Cancer Institute (https://www.cancer.gov) (R01CA187733 to J.K.R.; T32CA190216 to A.M.P.; F31CA247343 to J.T.R.; F32CA239436 to M.M.W.; P30CA046934 to the University of Colorado); the National Institute of Dental and Craniofacial Research (https://www.nidcr.nih.gov) (K99DE030528 to A.M.P.); the U.S. Department of Defense Congressionally Directed Medical Research Program Breast Cancer Research Program (https://cdmrp.army.mil/bcrp/) (BC170270 to A.M.P.); and the University of Colorado School of Medicine RNA Bioscience Initiative (https://medschool.cuanschutz.edu/rbi) (seed grant to A.M.J.). The funders had no role in study design, data collection and analysis, decision to publish, or preparation of the manuscript.

**Competing interests:** I have read the journal's policy and the authors of this manuscript have the following competing interests: A.M.J., A.M.P., and J.T.R are listed as co-inventors on a patent application related to this work (U.S. 63/187,835).

**Abbreviations:** DEG, differentially expressed gene; lncRNA, long noncoding RNA; LSD1, lysine-specific demethylase 1; m6A, N6-methyladenosine; PRC2, polycomb repressive complex 2; RIP, RNA immunoprecipitation; siRNA, small interfering RNA; TNBC, triple-negative breast cancer; WT, wild-type.

repress expression of the HoxD locus [5], aberrant high levels of HOTAIR are associated with poor survival and increased cancer metastasis in many different cancer types, including breast cancer [6,7]. Exogenous overexpression of HOTAIR in the MDA-MB-231 triple-negative breast cancer (TNBC) cell line results in the repression of hundreds of genes [6], and it promotes cell invasion, migration, proliferation, and self-renewal capacity in multiple breast cancer cell lines [6,8,9]. HOTAIR function is particularly striking in MDA-MB-231 cells, given that this is already a highly invasive breast cancer cell line, and its invasiveness is increased even further by HOTAIR overexpression [6,9]. This is reflective of the prognostic impact of HOTAIR expression in TNBC patients where high HOTAIR expression correlates with poorer overall survival [6,10]. MDA-MB-231 cells express low levels of endogenous HOTAIR, offering an opportunity to study response to HOTAIR transgenic overexpression, which is proposed to mimic the high levels of HOTAIR observed in patients with aggressive TNBC [6].

HOTAIR promotes polycomb repressive complex 2 (PRC2) activity, resulting in heterochromatin [6,11–14]. However, HOTAIR initially represses genes upstream of PRC2, though the mechanism is unclear [15] (Fig 1A). HOTAIR also interacts with the repressor lysine-specific demethylase 1 (LSD1) [12,16], although it was recently proposed to inhibit normal function of LSD1 in maintaining epithelial cells [17]. How HOTAIR specifically accomplishes initial transcriptional repression at its target loci, and how other pathways and cancer contexts influence HOTAIR function, remain elusive.

N6-methyladenosine (m6A) is a reversible RNA modification that can regulate multiple steps of the mRNA life cycle, including processing, decay, and translation [18]. How m6A regulates lncRNA-mediated processes is less understood. In one example, the lncRNA Xist, a key mediator of X chromosome inactivation, contains multiple m6A sites that contribute to its ability to induce repression of the X chromosome [19,20].

The m6A modification on an RNA is typically recognized by a "reader" protein that binds specifically to methylated adenosine to mediate the functional outcome of m6A deposition. Apart from the YTH family of proteins that contain the YTH domain that directly read m6A, a handful of non-canonical indirect m6A readers have been suggested [21]. In the case of Xist, the canonical YTH-containing nuclear localized m6A reader YTHDC1 recognizes m6A on Xist to mediate repression of the X chromosome [20,22]. In contrast, m6A on *cis*-acting chromatin-associated regulatory RNAs leads to their YTHDC1-dependent degradation, preventing transcription of downstream genes [23]. Collectively, m6A influences the regulatory roles of both mRNA and noncoding RNA via diverse mechanisms [24].

RNA modifications such as m6A have been shown to play critical roles in several human cancers [25]. In breast cancer, studies have revealed that dysregulation of m6A levels can generate breast cancer stem-like cells and promote metastasis [26–28]. Of the currently designated m6A reader proteins, we have previously shown that hnRNP A2/B1, a proposed non-canonical reader lacking the m6A-binding YTH domain, can interact with HOTAIR to regulate its chromatin and cancer biology mechanisms by promoting HOTAIR interactions with target mRNAs [9,14] (Fig 1A). Additionally, a recent proteomics screen found that components of the m6A methyltransferase complex WTAP and RBM15 bind HOTAIR [29]. This evidence suggests that m6A may play a role in cancers where HOTAIR is overexpressed.

Here, we investigate the function of m6A in HOTAIR-mediated breast cancer phenotypes. We identify 8 prominent m6A sites in HOTAIR and show that a single site (A783) is required for HOTAIR-mediated TNBC cell growth and invasion. We find that YTHDC1, the nuclear m6A reader, interacts with HOTAIR at methylated A783 and this interaction promotes chromatin association and gene repression upstream of PRC2. These results help explain why high HOTAIR is significantly associated with shorter overall patient survival, particularly in the context of high *YTHDC1*. Mutation of adenosine 783 in HOTAIR to uracil prevents nearly all

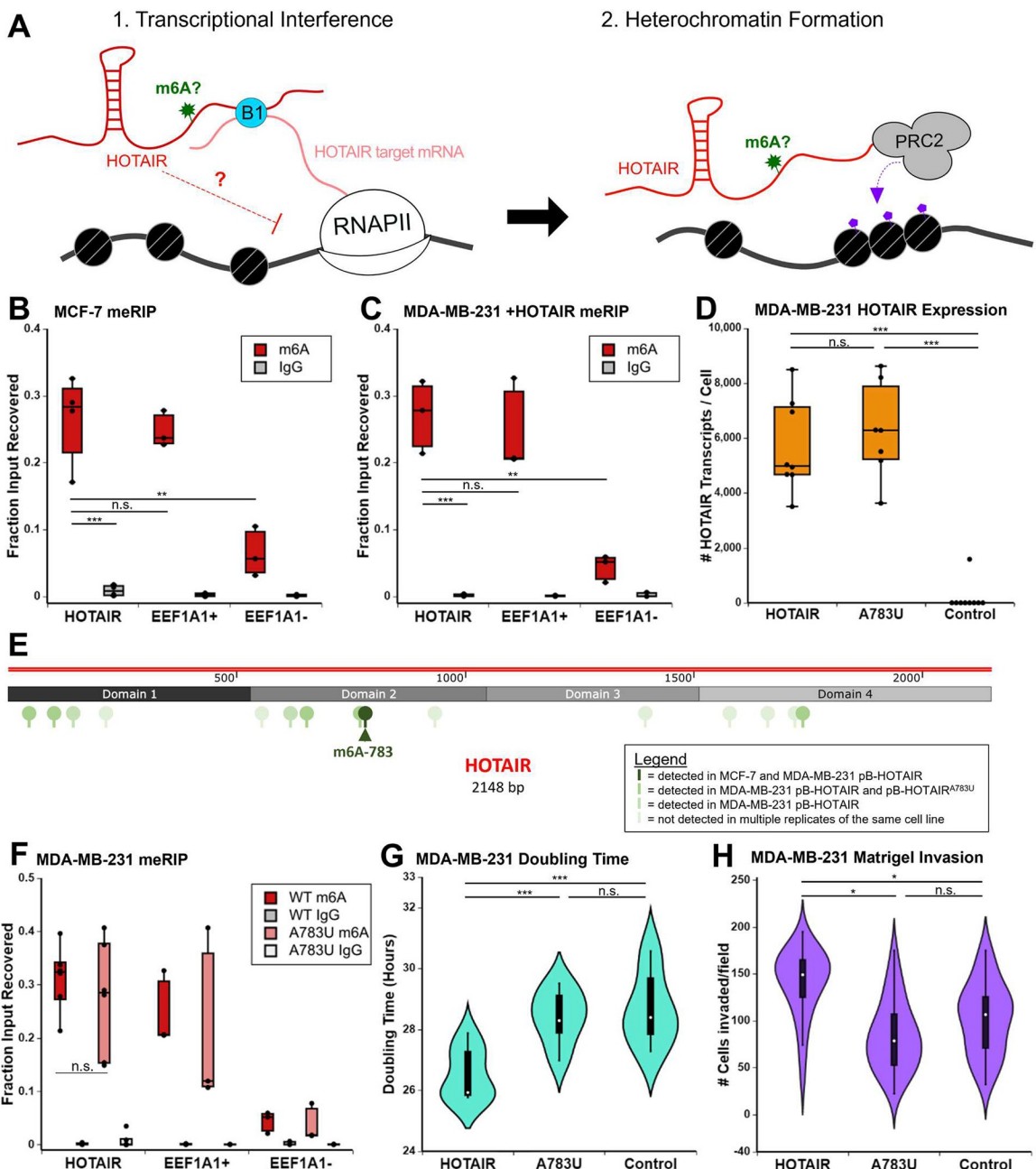

**Fig 1. LncRNA HOTAIR is m6A modified.** **(A)** General model for HOTAIR mechanism. HOTAIR is initially recruited to its target loci via RNA–RNA interactions with its mRNA targets that is mediated by hnRNP B1. HOTAIR association with chromatin induces transcriptional interference via an unknown mechanism, promoting heterochromatin formation by PRC2 through H3K27me3. This paper investigates the role of m6A on HOTAIR. **(B, C)** m6A RNA immunoprecipitation performed with an m6A antibody or IgG control in MCF-7 breast cancer cells (B) or MDA-MB-231 breast cancer cells with transgenic overexpression of HOTAIR (C). An m6A-modified region in EEF1A1 (EEF1A1 m6A) is a positive control, while a distal region in EEF1A1 that is not m6A modified (EEF1A1 distal) serves as a negative control. Three biological replicates are included. **(D)** Number of HOTAIR transcripts in MDA-MB-231 cells overexpressing WT HOTAIR, A783U HOTAIR, or an Anti-Luciferase control RNA. Experiments include 3 biological replicates each on 3 independently generated clones. **(E)** m6A sites detected in HOTAIR-expressing cells in 6 experiments (light green to dark green, scale of increasing occurrences). m6A site 783 (dark green, arrow) was detected in every experiment except where it was mutated. **(F)** m6A RNA immunoprecipitation performed with an m6A antibody or IgG control on MDA-MB-231 cells overexpressing WT HOTAIR, HOTAIR A783U, or an Anti-Luciferase control; 3–5 biological replicates were performed. **(G)** Doubling time of MDA-MB-231 overexpression cell lines described in (D). Experiments include 3 biological replicates each on 3 independently generated clones. **(H)** Quantification of Matrigel invasion assays performed with MDA-MB-231 overexpression cell lines described in (D). Four biological replicates were performed. Numerical values in panels 1B–D, 1F–H are included in S2 Data. lncRNA, long noncoding RNA; m6A, N6-methyladenosine.

of the normal gene expression changes that are induced by the WT lncRNA. Surprisingly, at a subset of genes, the A783U mutant induces opposite gene expression changes to WT HOTAIR, reducing molecular and cellular cancer phenotypes in TNBC cells, indicating that the mutant HOTAIR can function as an antimorph. Overall, our results suggest a model where a single site of m6A modification on HOTAIR enables a strong interaction with YTHDC1 for chromatin-mediated repression of its target genes, leading to altered TNBC properties. Collectively, our results demonstrate the potent activity of m6A on lncRNAs and in turn the role they play in cancer biology.

## Results

### HOTAIR contains multiple sites of m6A modification in breast cancer cell lines

To investigate the possibility that m6A regulates the function of HOTAIR in a mechanism similar to its regulation of lncRNA Xist, we examined previous genome-wide maps of m6A sites in human cells. Using the CVm6A database [30], we found 3 m6A peaks in HOTAIR in HeLa cells, although the enrichment score for these sites was low (S1A Fig). To evaluate m6A methylation of HOTAIR in relevant breast cancer cells, we performed m6A RNA immunoprecipitation (meRIP) qRT-PCR in MCF-7 cells, which express low levels of endogenous HOTAIR [9]. A significant portion of HOTAIR was recovered upon immunoprecipitation with the anti-m6A antibody (26.6 ± 6.6%, $p$ = 0.00024 versus IgG), similar to an m6A-modified region on the positive control region of *EEF1A1* (24.7 ± 2.7%, $p$ = 0.67), and consistently higher than a distal region of *EEF1A1* that is not m6A modified (6.4 ± 3.7%, $p$ = 0.0056) (Fig 1B).

We further found that m6A modification of HOTAIR is maintained during ectopic expression of HOTAIR in a stable MDA-MB-231 cell line. meRIP in MDA-MB-231 cells expressing transgenic HOTAIR resulted in significant HOTAIR recovery (27.1 ± 5.4%, $p$ = 0.001 versus IgG), similar to the EEF1A1 positive control (24.6 ± 7.0%, $p$ = 0.65) and significantly higher than the EEF1A1 negative control (4.4 ± 2.0%, $p$ = 0.02) (Fig 1C). As mentioned in the Introduction, this transgenic HOTAIR context is a model for high levels of HOTAIR observed in TNBC tumors. These results demonstrate that HOTAIR is m6A modified in 2 distinct breast cancer contexts representative of ER+ and TNBC.

To identify single nucleotide sites of m6A modification, we performed a modified m6A eCLIP protocol [31] on polyA-selected RNA from MCF-7 and MDA-MB-231 breast cancer cells (S1B Fig). In MCF-7 cells, we identified 1 m6A site within the HOTAIR transcript at adenosine 783 (S1 Table). Based on this and the RIP result in MCF-7s (Fig 1B), we estimate that A783 is approximately 25% methylated, at a minimum. m6A at adenosine 783 in MDA-MB-231 cells with transgenic HOTAIR was consistently detected with high confidence (S1 Table), along with 7 other sites using our multi-replicate consensus approach [31] (S2 Table). Of note, A783 occurred within a single-stranded region of the HOTAIR secondary structure [16] (S1C Fig) which promotes METTL3/14 dependent methylation [32]. To confirm m6A modification at adenosine 783 in MCF-7 cells, we performed additional meRIP experiments to quantify the level of m6A modification in specific regions of HOTAIR across its transcript, demonstrating that the region containing m6A783 (723–808) had the highest level of HOTAIR recovery (S2A Fig).

We performed additional m6A mapping experiments in three breast cancer cell lines that were recently generated from patient-derived xenografts [33]. One of these lines bears A783 m6A methylation as well, and this was the only m6A site identified in HOTAIR in these samples (S3 Table). HOTAIR levels were highest in this cell line (S3 Table and S2B Fig), suggesting that methylation status or detection ability scales with RNA level.

To test if HOTAIR is m6A modified by the canonical m6A methyltransferase METTL3/14 complex, we performed shRNA-mediated depletion of METTL3, METTL14, and the adaptor protein WTAP in MCF-7 cells (S2D Fig, left) where A783 is the only methylated residue detected. We observed approximately 3- to 5-fold reduced recovery of HOTAIR in methyltransferase-depleted cells relative to non-targeting controls ($p = 0.0063$) (S2D Fig, right). Together, these results indicate that m6A methylation of HOTAIR, particularly at A783, occur in breast cancer cells and is dependent on the METTL3/14 complex.

## Nucleotide A783 is important for the ability of HOTAIR to promote breast cancer cell proliferation and invasion

Given that nucleotide A783 was consistently methylated within HOTAIR in our m6A mapping experiments, in both endogenous and overexpressed contexts, we asked whether this modification had any consequences to HOTAIR function. To directly test the functional role of A783, we mutated the adenosine to uracil at this position (HOTAIR$^{A783U}$). Both wild-type (WT) and the mutant form of HOTAIR were expressed at similar levels averaging approximately 5,000 to 6,000 copies of HOTAIR per cell (Fig 1D), resembling the high levels of HOTAIR observed in samples from cancer patients [6,10,34]. We then mapped m6A sites in MDA-MB-231 cells overexpressing the HOTAIR$^{A783U}$ mutant as above (S1 Table). While the CLIP-based m6A signature was no longer detected at adenosine 783 when this site was mutated to uracil, we detected m6A modification at 5 of the 7 other multi-replicate consensus sites (S1 and S2 Tables). The results of the m6A mapping experiments in MCF-7 cells and MDA-MB-231 cells with transgenic overexpression of WT or A783U HOTAIR are summarized in Fig 1E. We note that nucleotides 143 and 620 were no longer called with multi-replicate consensus confidence as m6A in the A783U mutant, though m6A143 was only called in WT HOTAIR at our lowest confidence category and m6A620 is called in one of the A783U mutant replicates (S1 Table). Nonetheless, it is possible that methylation at A783 is required for one or both m6A events to occur. However, meRIP on MDA-MB-231 cells expressing HOTAIR$^{A783U}$ resulted in significant HOTAIR recovery ($27.6 \pm 10.8\%$, $p = 0.00012$ versus IgG), similar to WT HOTAIR-expressing cells ($31.2 \pm 6.1\%$, $p = 2.16 \times 10^{-7}$ versus IgG) (0.88-fold change, $p = 0.49$) (Fig 1F). meRIP analysis spanning the HOTAIR transcript in MDA-MB-231 cells expressing WT or A783U mutant HOTAIR also showed similar recovery across the HOTAIR transcript (S1D Fig). Recovery around A783 is unaffected by A783U, providing further evidence that the nearby m6A site at A772 remains methylated, in agreement with our meCLIP analysis (S1 Table). HOTAIR$^{A783U}$ had increased recovery in the region of HOTAIR containing m6A sites 143 and 215 (141–240) (A783U versus WT = 1.6, $p = 0.0009$) (S2D Fig). Altogether, these results suggest that HOTAIR$^{A783U}$ maintains m6A modification at other sites in HOTAIR.

To determine the effect of the A783U mutation on HOTAIR-mediated breast cancer cell growth, we measured the doubling time of MDA-MB-231 cells expressing WT and A783U mutant HOTAIR. As described above, we overexpressed HOTAIR and the HOTAIR$^{A783U}$ mutant in MDA-MB-231 cells and included overexpression of an antisense sequence of luciferase mRNA (Anti-Luc) as a negative control [9]. Similar to previous studies, transgenic overexpression of HOTAIR mediated increased cancer growth and invasion of MDA-MB-231 cells [6] (Fig 1G and 1H). We observed that MDA-MB-231 cells overexpressing WT HOTAIR proliferated more quickly, with a shorter doubling time ($26.5 \pm 0.83$ hours) than cells overexpressing Anti-Luc ($28.7 \pm 1.2$ hours, $p = 0.00046$) (Figs 1G and S2E). Surprisingly, the single nucleotide mutation of A783U in HOTAIR abolished its ability to enhance MDA-MB-231 cell proliferation; cells expressing HOTAIR$^{A783U}$ proliferated more slowly, with a longer doubling time than those expressing WT HOTAIR ($28.4 \pm 0.87$ hours, $p = 0.00026$) and grew similarly

to cells containing the Anti-Luc control ($p = 0.59$). To examine the role of A783 of HOTAIR in mediating breast cancer cell invasion, the same MDA-MB-231 cell lines were plated in a Matrigel invasion assay. Overexpression of WT HOTAIR induced a significant increase in number of cells invaded compared to the Anti-Luc control ($140.25 \pm 23$ in WT HOTAIR versus $102.7 \pm 16$ in control, $p = 0.038$). In contrast, overexpression of A783U HOTAIR did not lead to an increase in invasion compared to the Anti-Luc control ($84.1 \pm 22$, $p = 0.22$) and resulted in significantly less cells invaded compared to overexpression of WT HOTAIR ($p = 0.012$) (Fig 1H).

To confirm that the defects of the HOTAIR$^{A783U}$ mutant is due to lack of m6A methylation at that site and not due to an unintended structural alteration, we generated three additional HOTAIR mutants: A782U, A783C, and C784U (S3A Fig). Each of these mutations blocks m6A methylation at A783 either due to mutation of the m6A consensus sequence or by mutation of A783 itself, while having different local sequence changes than A783U. We generated stable MDA-MB-231 cell lines overexpressing each of these constructs and observed similar levels of HOTAIR expression across these lines (S3B Fig). Each mutation caused defects in HOTAIR-induced proliferation (S3C Fig) and invasion (S3D Fig), similarly to A783U. Altogether, these results suggest that m6A modification of adenosine 783 in HOTAIR is key for mediating the increased aggressiveness of TNBC that is promoted in contexts where the lncRNA is overexpressed.

## hnRNP B1 is not a direct m6A reader in MCF-7 cells

We next sought to address the mechanisms behind HOTAIR m6A783 function. hnRNP A2/B1 has previously been suggested to be a reader of m6A, and the B1 isoform has a high affinity for binding HOTAIR [9,35,36]. However, comparing our previously generated eCLIP results for hnRNP B1 [13] to the m6A eCLIP, both performed in MCF-7 cells, we found that, out of 10,470 m6A sites, only 417 (4%) were identified to contain an hnRNP B1 binding site within 1,000 nucleotides (S4A Fig). Upon mapping hnRNP B1 signal intensity relative to the nearby m6A site, we observed that hnRNP B1 is depleted directly over m6A sites (S4B Fig). These results suggest that hnRNP B1 is not a direct m6A reader, although m6A may indirectly promote its recruitment in some contexts. When comparing hnRNP B1 binding in HOTAIR with m6A sites, B1 binding peaks in MCF-7 cells occur in m6A-free regions of HOTAIR. Conversely, data from in vitro eCLIP analysis of B1 binding to unmodified HOTAIR reveal additional B1 binding peaks in Domain 1 of HOTAIR, one of which occurs near several m6A sites (S4C Fig). Altogether, these data suggest that m6A is not likely to directly recruit hnRNP B1 as a reader, although it could contribute to hnRNP B1 binding.

## YTHDC1 interacts with methylated A783 in HOTAIR

In light of the results for hnRNP A2/B1 described above, we turned to alternative candidate m6A readers of HOTAIR. YTHDC1 is a nuclear-localized m6A reader that binds m6A sites in noncoding RNAs, including the Xist lncRNA [20]. We reasoned that YTHDC1 was a strong candidate for interaction with HOTAIR, which is an lncRNA that is also primarily nuclear localized. To determine if YTHDC1 interacts with HOTAIR, we performed RNA immunoprecipitation (RIP) qRT-PCR using an antibody to YTHDC1. In both MCF-7 cells expressing endogenous HOTAIR where m6A783 is the only m6A site detected, and MDA-MB-231 cells overexpressing transgenic HOTAIR, a significant portion of HOTAIR RNA was recovered when using antibodies specific against YTHDC1 ($2.6 \pm 0.007\%$, $p = 0.003$ versus IgG; $17.4 \pm 9.3\%$, $p = 0.04$ versus IgG, respectively) (Fig 2A and 2B).

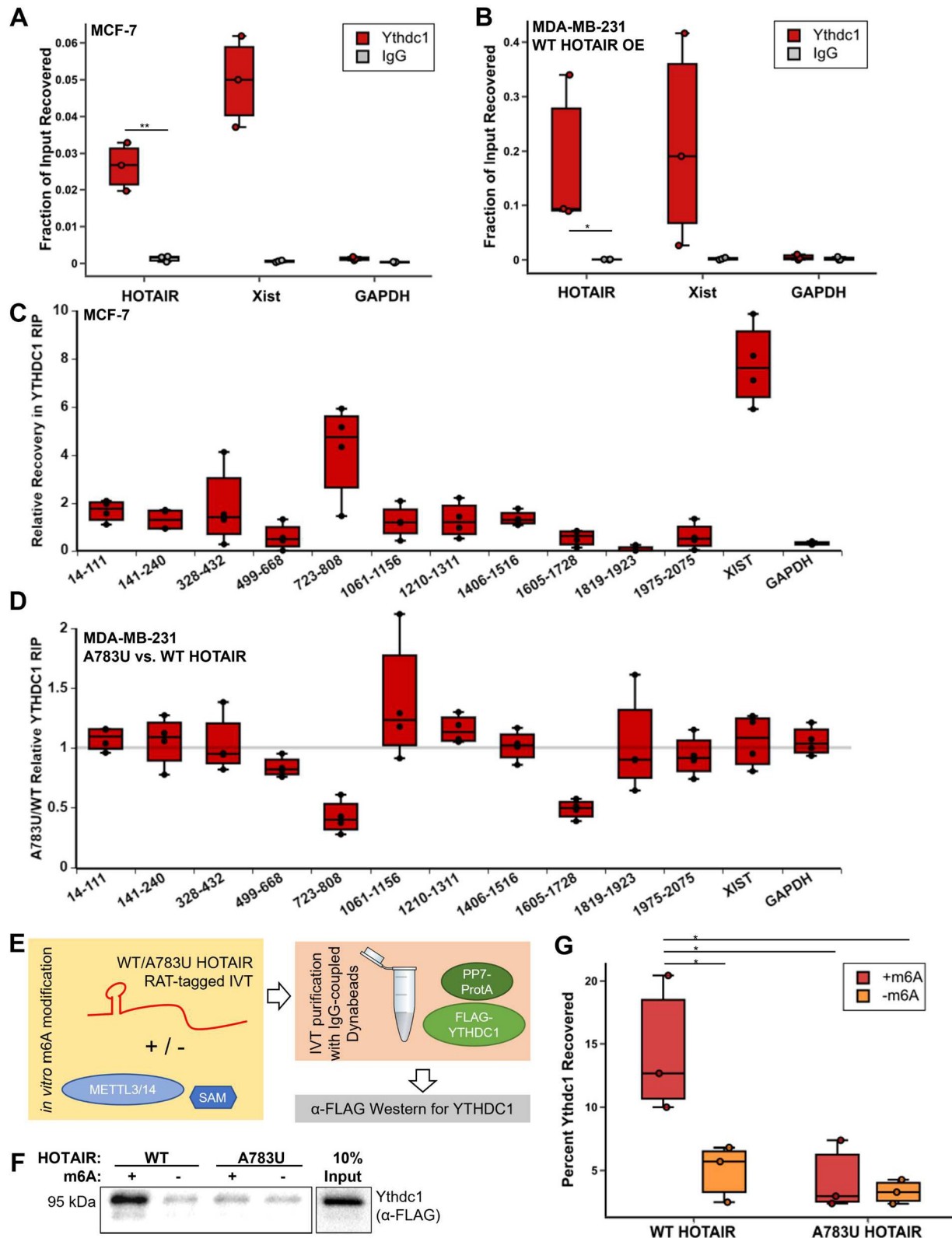

**Fig 2. YTHDC1 interacts with HOTAIR at m6A783. (A, B)** YTHDC1 RIP performed in MCF-7 cells (A) or MDA-MB-231 cells overexpressing transgenic HOTAIR (B) on 3 biological replicates. **(C)** YTHDC1 RIP performed in MCF-7 cells on 4 biological replicates. RNA recovery was monitored with qPCR probes noted in graph and normalized to recovery observed with qPCR oligos targeting a region of

HOTAIR with no detected m6A sites (1819–1923). **(D)** YTHDC1 RIP performed in MDA-MB-231 cells overexpressing WT HOTAIR or A783U HOTAIR. RNA recovery was monitored with qPCR as in (C) and fraction recovered in mutant compared to WT was determined. Four biological replicates were performed. **(E)** Schematic of YTHDC1 pulldown experiment with m6A-modified WT and A783U HOTAIR. PP7-tagged domain 2 of WT or A783U HOTAIR was in vitro transcribed and m6A modified with purified METTL3/14. RNA was bound to PP7-Protein A, cellular extract containing FLAG-tagged YTHDC1 was added, and a pulldown was performed with IgG-coupled Dynabeads. Amount of FLAG-YTHDC1 bound was assessed by western blot. **(F)** Anti-FLAG western blot of pulldown experiment outlined in (E). **(G)** Quantification of anti-FLAG western blots from 3 replicates. Numerical values in panels 2A–D, 2G are included in S2 Data. RIP, RNA immunoprecipitation; WT, wild-type.

To further examine interaction of YTHDC1 at m6A783, we performed YTHDC1 RIP experiments in MCF-7 cells, where A783 is the only methylated site detected, using qPCR probes targeting different regions of HOTAIR (see S4C Fig). We found that there was a significant enrichment in recovery of the region containing m6A783 compared to other regions of HOTAIR ($4.73 \pm 1.6$-fold increase versus 3′ region, $p = 0.0022$ versus IgG, $p \leq 0.02$ versus all other regions tested except 328–432 and Xist) (Fig 2C), supporting m6A-dependent association at A783. To determine whether m6A at A783 regulates interactions with YTHDC1, we performed YTHDC1 RIP experiments in MDA-MB-231 cells overexpressing WT or A783U mutant HOTAIR (Figs 2D and S4D). RNA recovery in the YTHDC1 immunoprecipitation in cells overexpressing WT HOTAIR was highest in the region containing A783 ($2.05 \pm 0.02$-fold increase versus 3′ region), but also observed across the HOTAIR transcript, suggesting the potential for other sites of YTHDC1 occupancy on the lncRNA in this context where we have observed additional m6A sites (S1 Table). While the majority of the HOTAIR transcript had similar recovery in WT versus A783U, region 723–808, the region containing the m6A783 site, was most affected by A783U mutation ($58 \pm 14\%$ reduced recovery in A783U versus WT, $p = 0.0003$), along with 1 additional region at 1605–1728 ($51 \pm 7.8\%$ reduced recovery in A783U versus WT, $p = 1.6 \times 10^{-5}$). This change in 1605–1723 is likely due to a secondary interaction of YTHDC1 with this region mediated by m6A783 rather than changes in direct m6A interactions, as we do not observe changes in m6A RIP recovery of this region in HOTAIR[A783U] (S2D Fig). These results demonstrate a defect in the HOTAIR[A783U] mutant in its ability to interact with YTHDC1, highlighting a loss of YTHDC1 at the region containing nucleotide 783 when it cannot be methylated.

To examine the association of other m6A readers, we performed similar RIP experiments using antibodies to YTHDF1 or YTHDF2 in MCF-7 cells. While a proportion of HOTAIR was recovered with these antibodies, there was no enrichment for the region containing m6A783 (S4E and S4F Fig). This supports a specific role for YTHDC1 as the m6A reader that targets m6A783 of HOTAIR.

To further confirm that nucleotide A783 in HOTAIR recruits YTHDC1 via m6A modification, we generated PP7-tagged in vitro transcribed RNA of domain 2 of WT or A783U mutant HOTAIR and performed in vitro m6A methylation with purified METTL3/14 [37] and S-adenosylmethionine as a methyl donor. As noted above, A783 has low structural propensity [16], making it a favorable METTL3/14 substrate. In this purified system, it is possible that other sites can be methylated, even those that do not occur in cells; however, in this context, the only difference between the 2 constructs used is the A783 base substitution, which prevents methylation at this site. The in vitro *HOTAIR* transcripts were tethered to magnetic beads and incubated with FLAG-YTHDC1-containing protein lysates, then the relative recovery of FLAG-YTHDC1 was determined by anti-FLAG western blot (Fig 2E). WT HOTAIR interaction with YTHDC1 was enhanced when the transcript was m6A modified (approximately 3-fold increase, $p = 0.04$), while A783U HOTAIR interaction with YTHDC1 was not significantly altered by the addition of m6A (approximately 1.3-fold change, $p = 0.6$) (Fig 2F and 2G). Further supporting a specific role for m6A783, we note the non-canonical "GAACG"

sequence at this location was identified as one of the top 10 most abundant m6A-centered 5-mer sequences to interact with YTHDC1 in an in vitro selection study [38]. Altogether, these results suggest that m6A783 of HOTAIR mediates a specific interaction with YTHDC1.

## YTHDC1 levels regulate HOTAIR-mediated proliferation of MDA-MB-231 cells and clinical outcomes

To test the role of YTHDC1 in HOTAIR's ability to enhance breast cancer cell proliferation, we stably overexpressed or knocked down YTHDC1 in the context of WT or A783U HOTAIR overexpression in MDA-MB-231 cells (Fig 3A and 3B). We noted that YTHDC1 protein levels tended to be approximately 2-fold higher in cells containing WT HOTAIR compared to A783U mutant HOTAIR (Fig 3B). Although this difference was not significant ($p$ = 0.16), it suggests a potential positive relationship between WT HOTAIR RNA and YTHDC1 protein levels, indicating that high levels of methylated A783 may stabilize a fraction of YTHDC1. Next, we used the MDA-MB-231 cell lines we generated to analyze proliferation as described above. Growth of MDA-MB-231 cells overexpressing WT HOTAIR was not significantly altered by YTHDC1 dosage (0.96-fold change, $p$ = 0.16 for pLX-DC1; 1.08-fold change, $p$ = 0.26 for shDC1, respectively), yet there was a trend towards decreased doubling time with increasing YTHDC1. In contrast, cells with A783U mutant HOTAIR had significant differences in doubling time with overexpression or knockdown of YTHDC1 (Fig 3C). Overexpression of YTHDC1 led to significantly faster growth of MDA-MB-231 cells containing A783U mutant HOTAIR (0.84-fold change in doubling time, $p$ = 0.003), with proliferation rates comparable to cells expressing WT HOTAIR. Knockdown of YTHDC1 in cells containing HOTAIR$^{A783U}$ was particularly potent in reducing the growth rate (approximately 1.2-fold increase in doubling time, $p$ = 0.008). These results suggest that without A783 methylation, the reduced occupancy of YTHDC1 specifically at A783 can be partially compensated by YTHDC1 overexpression and aggravated by knockdown. We suspect this could be mediated by the secondary m6A sites of HOTAIR, permitting some level of compensatory function. However, these results occur in the background of general manipulation of YTHDC1 levels, which likely has additional pleiotropic affects. We address this issue with more specifically targeted experiments using the dCasRX system (see section on "Tethering YTHDC1...").

To explore how breast cancer outcomes are affected by HOTAIR and YTHDC1 levels, we used Kaplan–Meier plotter [39]. We found that high HOTAIR levels were only significantly associated with decreased survival in the context of high YTHDC1 mRNA (S5A and S5B Fig), suggestive of a role for YTHDC1 in enhancing HOTAIR's ability to mediate more aggressive cancer. Interestingly, using UALCAN (a tool for analyzing cancer OMICS data) to determine gene expression in normal breast tissue versus breast tumor specimens [40], there appears to be a lack of concordance in the relationship between YTHDC1 mRNA and protein levels in clinical samples: Average YTHDC1 mRNA levels were decreased across a breast tumor panel, while average protein levels were increased (S5C and S5D Fig). However, we reason that most samples with high YTHDC1 mRNA levels were likely to have higher protein levels. These data hint that the cancer phenotypes dependent on HOTAIR association with YTHDC1 may have clinical implications. Discovery of methylated A783 in cells recently derived from a breast tumor (S3 Table) support this as well.

## m6A and YTHDC1 mediate chromatin association and expression of HOTAIR

Based on the differences observed between cell lines containing WT and A783U HOTAIR and the function of HOTAIR in chromatin-mediated gene repression, we investigated whether

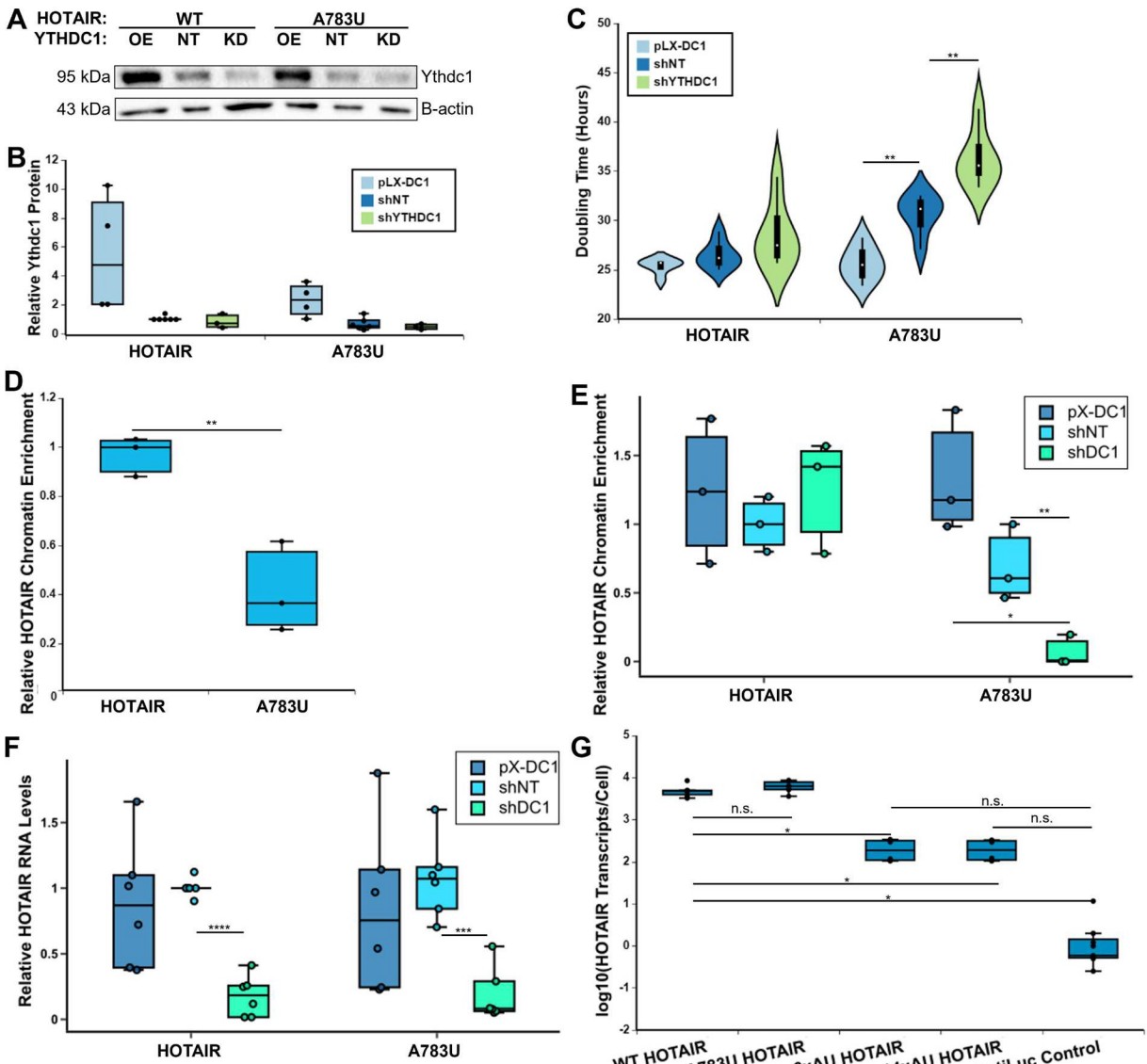

**Fig 3. YTHDC1 regulates HOTAIR activity and stability.** (A) Western blot results of YTHDC1 protein levels in pLX-DC1 overexpression, shNT control, and shDC1 knockdown MDA-MB-231 cell lines expressing WT or A783U HOTAIR. (B) Quantification of 3 replicates of (A). Protein levels of YTHDC1 were normalized to β-actin levels and are relative to the HOTAIR shNT sample. Four biological replicates were performed. (C) Doubling time of MDA-MB-231 cells containing WT or A783U HOTAIR and overexpression or knockdown of YTHDC1. Three biological replicates on 2 independently generated clones were performed. (D) qRT-PCR was performed on fractionated RNA samples from MDA-MB-231 cells containing overexpression of WT or A783U HOTAIR, and chromatin association was calculated by determining the relative chromatin-associated RNA to input and normalizing to 7SL levels and relative to WT HOTAIR samples. Three biological replicates were performed. (E) Chromatin enrichment was calculated similarly as in (D) in MDA-MB-231 cell lines expressing WT or A783U HOTAIR with knockdown or overexpression of YTHDC1. Values are relative to HOTAIR shNT samples. Three biological replicates each were performed on 2 independently generated clones. (F) qRT-PCR of HOTAIR RNA levels in MDA-MB-231 cell lines overexpressing WT or A783U HOTAIR containing overexpression or knockdown of YTHDC1. Three biological replicates each were performed on 2 independently generated clones. (G) qRT-PCR of *HOTAIR* RNA levels in MDA-MB-231 cell lines expressing WT, A783U, 6xAU, or 14xAU HOTAIR or an AntiLuc control. Three biological replicates each were performed on 2 independently generated clones. Numerical values in panels 3B–G are included in S2 Data. WT, wild-type.

chromatin association of HOTAIR was altered in these cells. We performed fractionation of MDA-MB-231 cells containing WT or A783U HOTAIR into cytoplasm, nucleoplasm, and chromatin fractions (S6A Fig, see Materials and methods). We isolated RNA from each

fraction and performed qRT-PCR for HOTAIR and GAPDH. Cells overexpressing WT HOTAIR had significantly more chromatin-associated HOTAIR (approximately 4.3-fold) than cells expressing A783U HOTAIR ($p < 0.05$) (Fig 3D), though overall levels of HOTAIR are unchanged (Fig 1E).

To examine the effect of YTHDC1 levels on HOTAIR chromatin association, we performed a similar fractionation experiment in MDA-MB-231 cells expressing WT or A783U HOTAIR with overexpression or partial knockdown of YTHDC1 (S6B Fig). While YTHDC1 levels did not significantly alter WT HOTAIR chromatin association, overexpression of YTHDC1 increased HOTAIR$^{A783U}$ chromatin association approximately 1.9-fold to similar levels as WT HOTAIR ($p = 0.05$), and knockdown resulted in a significant approximately 10-fold decrease in chromatin association ($p = 0.01$) (Fig 3E). A caveat to these analyses is that HOTAIR expression levels remained similar for DC1 overexpression lines compared to shNT control lines (0.8-fold change, $p = 0.3$), but were significantly decreased by approximately 5- to 10-fold in YTHDC1 knockdown lines for both WT and A783U mutant HOTAIR relative to shNT cell lines ($p = 3.24 \times 10^{-10}$) (Fig 3F). Hence, while DC1 knockdown has no effect on relative WT HOTAIR chromatin association, total HOTAIR RNA levels are decreased, including chromatin associated HOTAIR, which may explain a partial effect in Fig 3C. We reason that the differences observed between WT and A783U mutant HOTAIR are due to a high-affinity interaction of YTHDC1 with WT HOTAIR at m6A783 that enables chromatin association and is not affected by partial knockdown or overexpression. We note that YTHDC1 levels have a 40% reduction upon knockdown, leaving a reduced, but substantial, population for interaction with methylated A783 of WT HOTAIR. For A783U mutant HOTAIR that does not interact with YTHDC1 at this position, increasing the concentration of YTHDC1 can drive interaction at other m6A sites within the mutated HOTAIR, each of which has lower affinity for YTHDC1 and/or lower frequency of m6A modification, but on average results in at least 1 site being occupied. These interactions occur at a low level in cells with WT YTHDC1 levels and are most sensitive to knockdown of YTHDC1 (S6C Fig). Therefore, the A783U mutant, which only retains these proposed lower affinity sites and cannot properly compensate for loss of the critical A783 methylation, is particularly sensitive to YTHDC1 levels.

The fact that HOTAIR expression levels were significantly decreased in YTHDC1 knockdown lines for both WT and A783U mutant HOTAIR relative to shNT cell lines suggests that YTHDC1 regulates the expression or stability of HOTAIR, independently of A783. This is consistent with YTHDC1 RIP recovery in other regions of HOTAIR in addition to A783 (S4D Fig). To investigate the role of other m6A sites within HOTAIR, we generated HOTAIR overexpression constructs containing 6 or 14 adenosine-to-uracil mutations (6xAU and 14xAU, respectively) both of which included A783U. While WT and A783U HOTAIR expression levels were similarly high, there was approximately 50-fold decrease in expression of 6xAU or 14xAU HOTAIR (Fig 3G). This suggests that other m6A sites within HOTAIR mediate its high expression levels in breast cancer cells. As expected, due to their decreased HOTAIR expression levels, we also observed longer doubling times for the multiple m6A HOTAIR mutants, similar to the single A783U mutant (S6D Fig).

## YTHDC1 contributes to gene repression by HOTAIR in the absence of PRC2, independent of its role in chromatin association or RNA stability

As HOTAIR induces repression of its direct target genes [5], we determined the effect of YTHDC1 on transcriptional repression mediated by HOTAIR. Using previously generated reporter cell lines that contain HOTAIR artificially directly tethered to chromatin upstream of a luciferase reporter gene to repress expression, independent of PRC2 [15] (Fig 4A), we

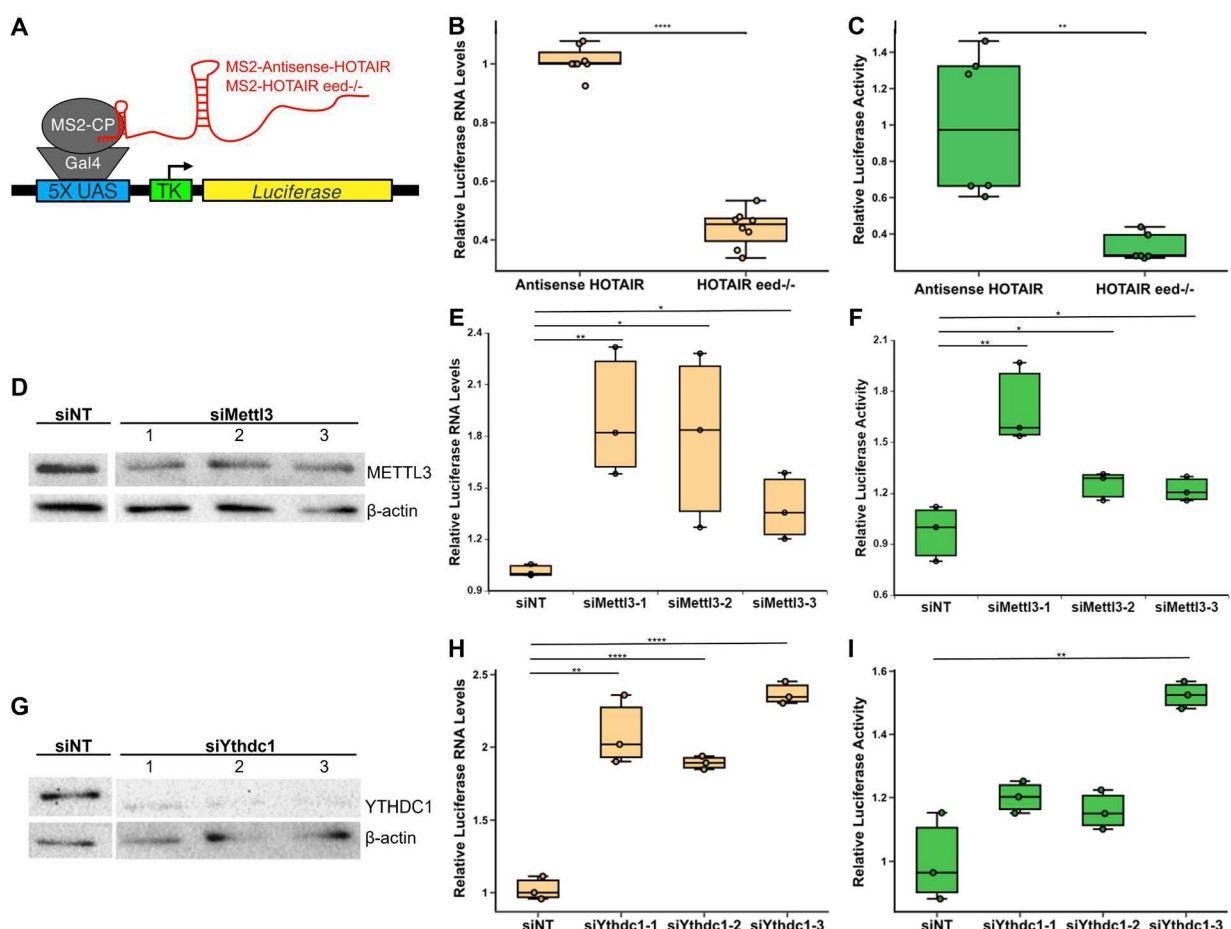

**Fig 4. YTHDC1 mediates transcriptional repression by HOTAIR. (A)** Schematic of 293T cells containing MS2-Antisense-HOTAIR or MS2-HOTAIR tethered upstream of a luciferase reporter. MS2-HOTAIR tethered cells also contain a deletion of *EED*, a subunit of PRC2 that is critical for H3K27 methylation. **(B, C)** Relative luciferase RNA levels (B) and relative luciferase activity (C) in MS2-Antisense HOTAIR or MS2-HOTAIR eed-/- cell lines. Six biological replicates were performed. **(D)** Western blot of METTL3 in MS2-HOTAIR eed-/- 293T reporter cells transfected with non-targeting siRNA or 3 different siRNAs targeting METTL3. **(E, F)** Relative luciferase RNA levels (E) and relative luciferase activity (F) of HOTAIR-tethered eed-/- cells transfected with a non-targeting siRNA or 3 different siRNAs against METTL3. Three biological replicates were performed for each experiment. **(G)** Western blot of YTHDC1 in MS2-HOTAIR eed-/- 293T reporter cells transfected with non-targeting siRNA or 3 different siRNAs targeting YTHDC1. **(H, I)** Relative luciferase RNA levels (E) and relative luciferase activity (F) of HOTAIR-tethered eed-/- cells transfected with a non-targeting siRNA or 3 different siRNAs against YTHDC1. Three biological replicates were performed for each experiment. Numerical values in panels 4B–C, 4E–F, 4H–I are included in S2 Data. siRNA, small interfering RNA.

confirmed that HOTAIR tethered upstream of the luciferase reporter reduced luciferase expression using both qRT-PCR (approximately 2.3-fold lower, $p = 8.0 \times 10^{-12}$) and luciferase assay (approximately 3.1-fold lower, $p = 0.002$) (Fig 4B and 4C). We also performed m6A eCLIP to confirm that *HOTAIR* was m6A modified in this context and detected 10 m6A sites within *HOTAIR*, including A783 (S1 Table). To test the role of m6A and YTHDC1 in the repression mediated by HOTAIR, we used 3 different small interfering RNAs (siRNAs) to knock down METTL3 or YTHDC1 relative to a non-targeting control (approximately 1.5-fold decrease in METTL3 protein levels, $p = 0.04$; approximately 2-fold decrease in YTHDC1 protein levels, $p = 0.02$) in the HOTAIR-tethered cells lacking the essential PRC2 subunit EED (Fig 4D and 4G). Knockdown of METTL3 or YTHDC1 resulted in significantly higher luciferase RNA levels in these cells (approximately 1.7-fold change for METTL3, $p = 0.0003$; approximately 2.1-fold change for YTHDC1, $p = 2.2 \times 10^{-5}$) (Fig 4E and 4H). Luciferase enzymatic

activity also increased upon knockdown of either METTL3 or YTHDC1 (approximately 1.4-fold change for METTL3, $p$ = 0.01; 1.3-fold change for YTHDC1, $p$ = 0.03) (Fig 4F and 4I). We note that knockdown did not affect *HOTAIR* RNA levels in this context (S7A and S7B Fig), indicating that the effects observed on luciferase expression were likely due to disruption of the HOTAIR gene repression mechanism via depletion of METTL3 or YTHDC1 protein rather than loss of HOTAIR expression (S7C Fig).

## Overexpression of A783U mutant HOTAIR induces divergent gene expression changes from wild-type HOTAIR in breast cancer cells

The connections we established between m6A and HOTAIR-dependent chromatin regulation prompted us to determine how widespread this was and whether it could explain the effects on breast cancer cell phenotype we observed when disrupting m6A783. To analyze HOTAIR-mediated gene expression changes in MDA-MB-231 cells, we performed high-throughput RNA sequencing on cells overexpressing WT HOTAIR, A783U mutant HOTAIR, or antisense luciferase as a control. For cells expressing WT HOTAIR, we identified 155 genes that were differentially expressed (adjusted $p$ < 0.1) when compared with control cells expressing Anti-Luc (Fig 5A and S1 Data). Down-regulated genes in cells expressing WT HOTAIR include genes involved in cell adhesion ($p$ = 0.0118), p53 ($p$ = 0.0112) and MAPK ($p$ = 0.0313) signaling, and tumor suppressors such as HIC1 and DMNT3A. Up-regulated genes with WT HOTAIR, generally presumed to be due to secondary effects from genes directly repressed by HOTAIR, include genes involved in positive regulation of angiogenesis ($p$ = $1.22 \times 10^{-5}$), regulation of cell population proliferation ($p$ = 0.0361), and cell differentiation ($p$ = 0.0157). As expected based on the role of YTHDC1 in HOTAIR-mediated gene repression (Fig 4H and 4I), the A783U mutant was unable to repress nearly all genes that WT HOTAIR could (Fig 5A).

Surprisingly, mutation of A783 did not merely prevent most gene expression changes seen in WT HOTAIR, but instead, expression of the A783U mutant induced certain changes in the opposite direction from the baseline control MDA-MB-231 cell line. We confirmed this pattern by qRT-PCR: genes down-regulated with WT HOTAIR were significantly increased with A783U HOTAIR, compared to the parental MDA-MB-231 control, including *SEMA5A*, a guidance cue protein that suppresses the proliferation and migration of cancer cells [41,42] (fold change relative to control in WT HOTAIR = −3.5, $p$ = 0.0002; in A783U HOTAIR = 2.8, $p$ = 0.01); *SIRPA*, a cell surface receptor that can act as a negative regulator of the phosphatidylinositol 3-kinase signaling and MAPK pathways [43] (fold change in WT HOTAIR = −1.6, $p$ = 0.03; in A783U HOTAIR = 3.1, $p$ = 0.009); and *TP53I11*, a p53-interacting protein that suppresses migration and metastasis in MDA-MB-231 cells [44] (fold change in WT HOTAIR = −1.7, $p$ = 0.03; in A783U HOTAIR = 4.0, $p$ = 0.008) (Fig 5B). Similarly, genes that were up-regulated in MDA-MB-231 cells upon introduction of WT HOTAIR had decreased expression in cells with A783U mutant HOTAIR (Fig 5C). This included genes such as *PTK7*, a transforming gene and prognostic marker for breast cancer and nodal metastasis involvement [45] (fold change relative to control in WT HOTAIR = 2.6, $p$ = 0.008; in A783U HOTAIR = −1.4, $p$ = 0.002); *CDH11*, a mesenchymal cadherin that is up-regulated in invasive breast cancer cell lines [46] (fold change in WT HOTAIR = 2.0, $p$ = 0.01; in A783U HOTAIR = −1.6, $p$ = 0.02); and *GRIN2A*, an oncogenic glutamate receptor (fold change in WT HOTAIR = 2.5, $p$ = 0.006; in A783U HOTAIR = −3.7, $p$ = $9.3 \times 10^{-5}$). There is evidence that these genes are directly targeted by HOTAIR, including a HOTAIR–chromatin interaction in SEMA5A which was detected by ChIRP [47], and HOTAIR-dependent H3K27me3 peaks in TP53I11 and SIRPA detected by ChIP [9]. Interestingly, the majority of genes that are differentially expressed in

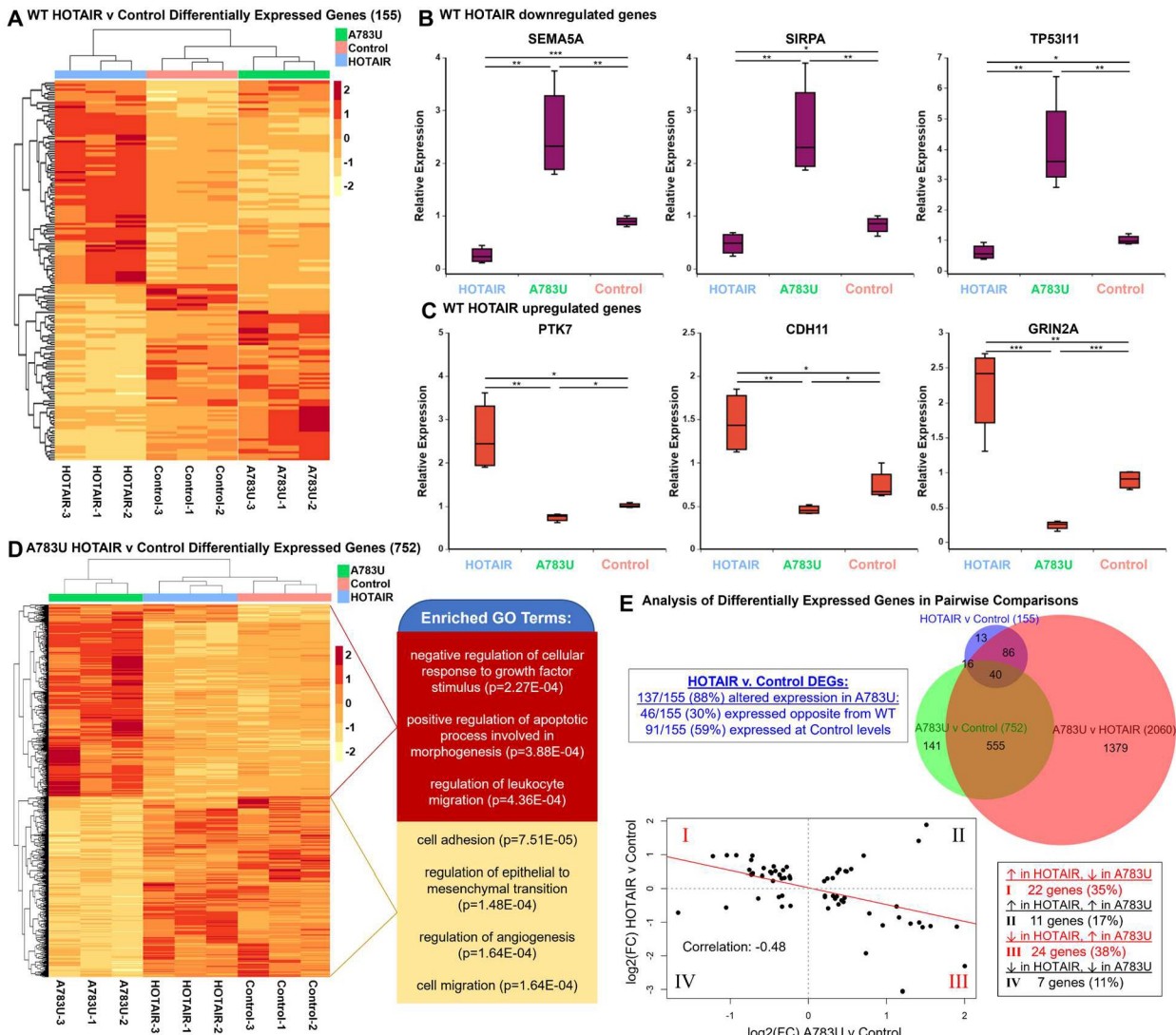

**Fig 5. HOTAIR-mediated gene expression changes in breast cancer are altered by mutation of A783. (A)** Heat map of Z-scores of DEGs between MDA-MB-231 cells overexpressing WT HOTAIR versus an Anti-Luciferase control. **(B, C)** qRT-PCR analysis of genes up-regulated (B) or down-regulated (C) upon HOTAIR overexpression. **(D)** Heat map of Z-scores of DEGs between MDA-MB-231 cells overexpressing A783U mutant HOTAIR versus an Anti-Luciferase control, left. Selected significant GO terms in up-regulated (red) and down-regulated (yellow) genes, right. **(E)** Additional analysis of DEGs. Top, Venn diagram (created using BioVenn [66]) of number of DEGs between MDA-MB-231 cells overexpressing WT HOTAIR, A783U mutant HOTAIR, or an Anti-Luciferase control. Top left inset describes the direction of change in A783U v. Control relative to the direction of WT HOTAIR v. Control, based on adjusted $p < 0.1$. Bottom, correlation analysis of expression in HOTAIR vs. Control and A783U vs. Control pairwise comparisons. Linear regression was used to fit a trend line (red) over the points representing DEGs in HOTAIR vs. Control, Pearson correlation coefficient included. Bottom right inset describes the number of genes in each quadrant. Numerical values in panels 5B–C, 5E are included in S2 Data. DEG, differentially expressed gene; WT, wild-type.

cells overexpressing HOTAIR versus control cells that bear HOTAIR ChIRP peaks (65%) or that have HOTAIR-dependent H3K27me3 (81.25%) display an antimorph pattern of expression (S1 Data). While there are likely some indirect gene regulatory events occurring, it appears that the majority of the genes that intersect in this antimorph mechanism are direct targets of the repressive function of HOTAIR.

To further analyze differences in cells expressing A783U mutant HOTAIR, we performed a pairwise comparison with control MDA-MB-231 cells and identified 758 differentially

expressed genes (DEGs) (Fig 5D and S1 Data). Up-regulated gene categories in A783U HOTAIR-expressing cells include negative regulation of response to growth factor stimulus ($p = 2.27 \times 10^{-4}$), positive regulation of apoptosis ($p = 3.88 \times 10^{-4}$), and regulation of migration ($p = 4.36 \times 10^{-4}$), while down-regulated gene categories include regulation of the epithelial to mesenchymal transition ($p = 1.48 \times 10^{-4}$), angiogenesis ($p = 1.64 \times 10^{-4}$), cell adhesion ($p = 7.51 \times 10^{-5}$), and cell migration ($p = 1.64 \times 10^{-4}$) (Fig 5D). We hypothesize that this altered pattern of gene expression may underlie the slight decrease in cell invasion observed in the A783U context compared to control MDA-MB-231 cells (Fig 1H). We also performed pairwise comparison of differences between cells expressing WT HOTAIR versus the A783U mutant HOTAIR. Here, we observed the most differentially expressed genes (2060) compared to other pairwise comparisons (Fig 5E, top, S1 Data).

Overall, these results reveal that expression of the A783U mutant HOTAIR induces additional and often opposite gene expression changes compared to expression of WT HOTAIR in breast cancer cells, suggesting a potential antimorph property of this single nucleotide mutation. The opposite gene expression pattern is evident in the heat map of all DEGs (S8A Fig), as well as the observation that most (137/155, 88%) of WT HOTAIR-regulated genes have altered expression with A783U HOTAIR, with a significant portion (46/155, 30%) having opposite expression, not simply loss-of-function, in MDA-MB-231 cells expressing A783U HOTAIR compared to control cells (Figs 5E and S8B and S8C). This pattern is also evident in the negative correlation (−0.48) when fold change in expression for HOTAIR v Control and A783U v Control is plotted (Fig 5E, bottom). We hypothesize that blocking m6A methylation at A783 prevents YTHDC1 interaction specifically at this site, which ultimately disrupts YTHDC1 function in stabilization of chromatin association and gene repression, leading to loss-of-function and antimorph cell biology and gene expression behaviors.

## Tethering YTHDC1 to A783U mutant HOTAIR restores chromatin association, proliferation, invasion, and HOTAIR-mediated gene repression

While overexpression and knockdown of YTHDC1 revealed effects on HOTAIR chromatin association (Fig 3E) and promotion of cell proliferation (Fig 3C), we recognize the potential caveat that general changes to YTHDC1 levels will not only affect the HOTAIR mechanism and that pleiotropic effects are possible. To more directly examine the effects of YTHDC1 interaction specifically with HOTAIR at A783, we employed a catalytically inactive RNA-targeting Cas protein, dCasRX, which has previously been used to recruit effectors to specific RNA molecules via a guide RNA (Fig 6A) [48]. We transfected MDA-MB-231 cells stably expressing WT or A783U HOTAIR with a plasmid containing the dCasRX-YTHDC1 fusion protein, in combination with a plasmid containing either a HOTAIR guide RNA (targeting a 22-nucleotide sequence 7 nucleotides downstream from A783 in HOTAIR, see Fig 6A) or a non-targeting gRNA. Expression of dCasRX-YTHDC1 was confirmed by western blot (Fig 6B). While chromatin association of WT HOTAIR remained consistently high, chromatin association levels of A783U HOTAIR were only restored to near WT HOTAIR levels upon transfection with plasmids containing the dCasRX-YTHDC1 fusion protein and the HOTAIR gRNA ($p = 0.25$ versus WT HOTAIR). In contrast, chromatin association of A783U HOTAIR remained low upon transfection of dCasRX-YTHDC1 with a non-targeting guide RNA (appropriately 3.7-fold lower than WT HOTAIR, $p = 0.0066$) (Fig 6C). HOTAIR RNA levels remained consistent in all samples (Fig 6D). These results confirm that YTHDC1 mediates chromatin localization of HOTAIR and show that the chromatin association defect of the A783U mutation can be restored simply by restoring binding of YTHDC1 at that specific

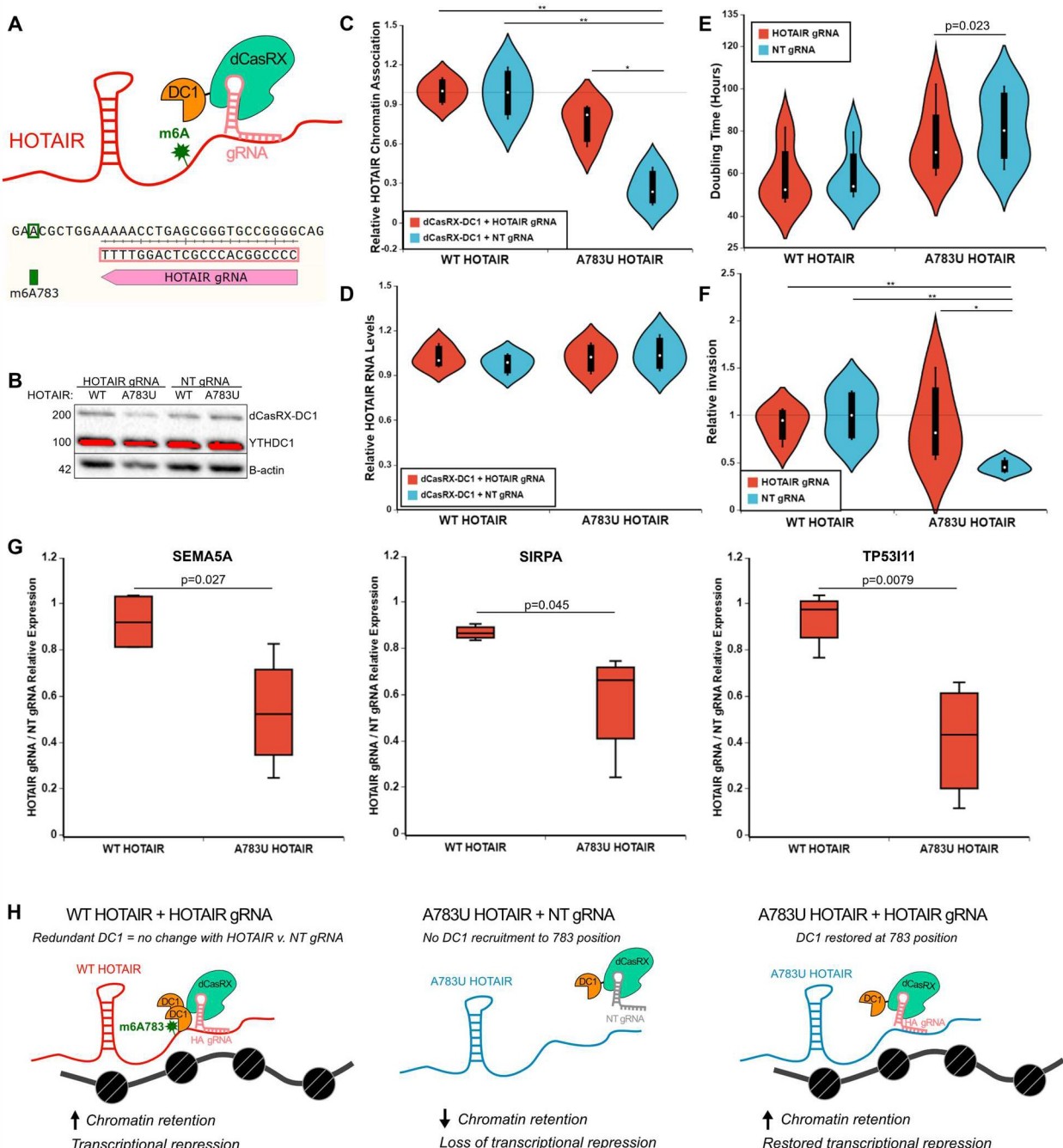

**Fig 6. Tethering YTHDC1 to A783U mutant HOTAIR restores chromatin localization independent of changes in RNA levels.** (**A**) Schematic of tethering strategy using a dCasRX-YTHDC1 fusion protein and a guide RNA targeted just downstream of A783 in HOTAIR. (**B**) Western blots for YTHDC1 (upper) and B-actin (lower) on dCasRX-YTHDC1 transfected cells, as noted. (**C**) Similar analysis described in Fig 3D and 3E was performed on fractionated RNA samples from cell lines overexpressing WT or A783U HOTAIR transfected with a plasmid containing dCasRX-YTHDC1 in combination with a HOTAIR or NT gRNA, as noted. Three biological replicates were performed. (**D**) Relative HOTAIR RNA levels in Input samples from C. Three biological replicates were performed. (**E**) Doubling time of samples described in C. Four biological replicates were performed. (**F**) Relative Matrigel invasion of samples described in C was determined. Six biological replicates were performed. Samples that did not show higher invasion in WT HOTAIR + NT gRNA vs. A783U HOTAIR + NT gRNA were excluded from the analysis. Invasion was normalized to WT HOTAIR + NT gRNA samples. Final analysis represents 4 biological replicates. (**G**) Relative expression of HOTAIR gRNA samples compared to NT gRNA samples for WT HOTAIR-down-regulated genes noted. (**H**) Schematic for observed effects of tethering YTHDC1 to HOTAIR using dCasRX. Numerical values in panels 6C–G are included in S2 Data. NT, non-targeting; WT, wild-type.

location. We also found that doubling time (Fig 6E) and invasion (Fig 6F) were partially restored upon tethering of YTHDC1 to A783U mutant HOTAIR, further supporting YTHDC1's role in mediating these effects.

To examine changes in HOTAIR-mediated gene expression changes upon tethering of YTHDC1 to A783U HOTAIR, we performed qRT-PCR on the same transfections described above. Tethering YTHDC1 to WT HOTAIR did not result in changes in gene expression of the HOTAIR-repressed genes we examined, presumably because WT HOTAIR is already bound by YTHDC1 at m6A783, so tethering additional YTHDC1 is redundant (Fig 6G and 6H). However, tethering YTHDC1 to A783U HOTAIR resulted in reduced expression of these genes (Fig 6G), suggesting a partial restoration of WT HOTAIR regulation. Interestingly, genes up-regulated with WT HOTAIR did not reveal significant changes in expression upon YTHDC1 tethering to A783U HOTAIR (S8D Fig). Together, our results support a mechanism where YTHDC1 interaction specifically at m6A783 leads to repression of HOTAIR genomic targets, and that this direct action causes further gene expression changes, cumulatively promoting HOTAIR- and m6A-dependent cancer cell phenotypes.

## Discussion

Similar to m6A regulation of mRNAs, it is becoming evident that m6A on lncRNAs is both functionally diverse and context dependent. Here, we demonstrate that m6A and the m6A reader YTHDC1 function to enable transcriptional repression by HOTAIR that is analogous to repressive functions demonstrated for the lncRNA Xist [20,22]. Our results reveal a mechanism whereby m6A modification of HOTAIR at a specific adenosine residue mediates interaction with YTHDC1, enabling transcriptional interference by HOTAIR that enhances TNBC properties including proliferation and invasion.

### Function of specific m6A sites in HOTAIR

While several m6A sites were identified within HOTAIR when overexpressed, we only detected 1 m6A site in the endogenously expressed context in MCF-7 and UCD4 cell lines, making it the most consistently present methylation site. It appears that HOTAIR methylation generally scales with its expression level, as we only detected the single site of m6A783 in endogenous HOTAIR in cell lines where it was more highly expressed and additional sites of m6A were only identified when HOTAIR was overexpressed in our transgene model. MDA-MB-231 cells overexpressing HOTAIR containing a mutation of this single m6A-modified adenosine had a defect in HOTAIR-mediated proliferation and invasion, as well as its ability to induce HOTAIR-mediated gene expression changes. Our results lead us to conclude that these changes are mediated through lack of methylation at this single site. This is reinforced by additional mutations in HOTAIR that prevent m6A methylation at A783 which also block HOTAIR-induced proliferation and invasion of TNBC cells, as well as the ability to restore WT activity by tethering the m6A reader YTHDC1 to A783U HOTAIR, bypassing the need for m6A modification at that site. It is likely that the A783 m6A site specifically is a high-affinity site for YTHDC1 interaction based on our in vivo analyses that found increased interaction with the region of HOTAIR containing A783 and reduced YTHDC1 binding upon mutation of A783, as well as the in vitro analysis where YTHDC1 association with domain 2 of HOTAIR was dependent on methylation of this site (Fig 2D and 2G). In addition, partial knockdown of YTHDC1 affects the stability of WT HOTAIR promoted by the secondary sites, but not its A783-related functions (Figs 3E and S6), suggesting that even with lower YTHDC1 levels, m6A783 is still bound by YTHDC1 while secondary sites do not have high enough affinity to maintain association.

While it is evident that m6A modification of A783 in HOTAIR is important for mediating its effects in breast cancer, other m6A sites within HOTAIR appear to play a role in enabling its high expression levels, potentially through transcript stabilization. When we bypass the normal mechanism of chromatin association using a direct tethering approach for HOTAIR (Fig 4A) [15], YTHDC1 is no longer required for chromatin association or stability, yet is required for gene repression, suggesting a direct role in shutting down transcription, perhaps with LSD1 involvement [12,17]. Our work emphasizes the importance of studying the function of individual m6A sites, as each m6A site has the potential to contribute to the function of an RNA in different ways. While our work is limited by the use of established laboratory cell lines, we note potential for therapeutic targeting of m6A783, based on the detection of this site in a recently patient-derived cell line, indicating the potential for its presence in patient tumor cells.

## m6A in gene repression and heterochromatin formation

HOTAIR and other lncRNAs make many dynamic and multivalent interactions with proteins that interact with other proteins, RNA molecules, and chromatin. In the nucleus, the METTL3/14 complex and YTHDC1 are key interactors with m6A-modified RNA that have been shown to regulate chromatin. Work in mouse embryonic stem cells has shown that METTL3 interacts with the SETD1B histone modifying complex, and this plays a role in repression of specific families of endogenous retroviruses [49]. However, due to the nature of HOTAIR's mechanism of repressing genes in *trans*, it is unlikely that the METTL3/14 complex remains bound to HOTAIR to induce repression of target loci. For YTHDC1, recent work has found that RNA interactions with this protein can directly regulate chromatin via recruitment of KDM3B, promoting H3K9me2 demethylation and gene expression [50]. In contrast, this study demonstrates that YTHDC1 can act to regulate chromatin association and transcriptional repression by HOTAIR, although the precise mechanism by which this is accomplished remains to be determined. Our data suggest that YTHDC1-mediated transcriptional repression occurs upstream of chromatin modification by PRC2. This supports the mechanism of transcriptional interference by HOTAIR proposed by Portoso and colleagues [15] (Fig 1A) and suggests that YTHDC1 is an important factor that mediates repression by HOTAIR. We provide evidence that PRC2 can act at many of the loci that we profile in the current study, and that these targets also display antimorph behavior when m6A783 is mutated. Interestingly, these include the genes that are transcriptionally down-regulated by HOTAIR overexpression and those that are up-regulated (S1 Data). This is informative for how a repressive lncRNA can cause up-regulation of some genes, likely through repression of a repressor locus near that gene. Open questions remain, however, as to how YTHDC1 binding to a repressive lncRNA mediates transcriptional interference and repression, which further studies will be needed to determine.

## Divergence in m6A and YTHDC1 function for different classes of RNAs

The role of YTHDC1 in mediating chromatin association of and repression by HOTAIR is interesting in the context of the recently identified broad nuclear role of YTHDC1 in regulation of transcription and chromatin state in mouse embryonic stem cells [23]. While in this case, it was demonstrated that YTHDC1 mediates degradation of m6A-modified chromatin-associated regulatory RNAs; our work raises the possibility that YTHDC1 might also mediate transcriptional repression and/or heterochromatin directly through interaction with regulatory RNAs. Our work also shows that, rather than degradation of HOTAIR, m6A sites in HOTAIR mediate its high expression in breast cancer via YTHDC1. The ability of YTHDC1

to form phase separated nuclear condensates, which has been shown to stabilize m6A-modified RNAs in myeloid leukemia cells [51], may play a role in stabilizing HOTAIR, providing a large reservoir that leads to aberrant targeting of genomic loci in cancer.

We hypothesize that chromatin association of HOTAIR stabilizes it because stable retention of HOTAIR on chromatin as heterochromatin forms is likely to make it inaccessible to factors that mediate its degradation. Our experiments where HOTAIR is tethered to chromatin in a reporter cell line illustrates this, as knockdown of YTHDC1 did not alter the stability of HOTAIR in the context where it is constitutively tethered to chromatin (S7C Fig). It is likely that YTHDC1 performs multiple functions within the nucleus, and that its effects on its target RNAs are context dependent, such as on other nearby RNA-binding proteins and/or local chromatin state.

Our work also highlights the fate of HOTAIR-YTHDC1 interaction that is distinctly different from mRNAs whose nuclear export is mediated by YTHDC1 [52]. In contrast, we show that YTHDC1 mediates chromatin association of the primarily nuclear-localized HOTAIR lncRNA. Also, while one specific m6A site at A783 is important for mediating chromatin association and the physiological effects of HOTAIR in breast cancer, other m6A sites play a role in overall expression or stability (Fig 7). Our results suggest that YTHDC1 and HOTAIR stabilize each other, as cells expressing WT HOTAIR have increased YTHDC1 protein levels (Fig 3A). A possible mechanism could be via stabilized phase-separated nuclear bodies analogous to nuclear YTHDC1-m6A condensates in leukemia [51]. These nuclear condensates stabilize m6A-modified mRNAs to enable their expression in AML cells; similar condensates could be linked to how YTHDC1 and HOTAIR stabilize each other. We propose this could also contribute to the mechanism of repression by HOTAIR, as phase separation can mediate heterochromatin formation [53] and has been proposed to contribute to X inactivation by Xist [54]. It is likely that the RNA context and other proteins that either interact directly with YTHDC1,

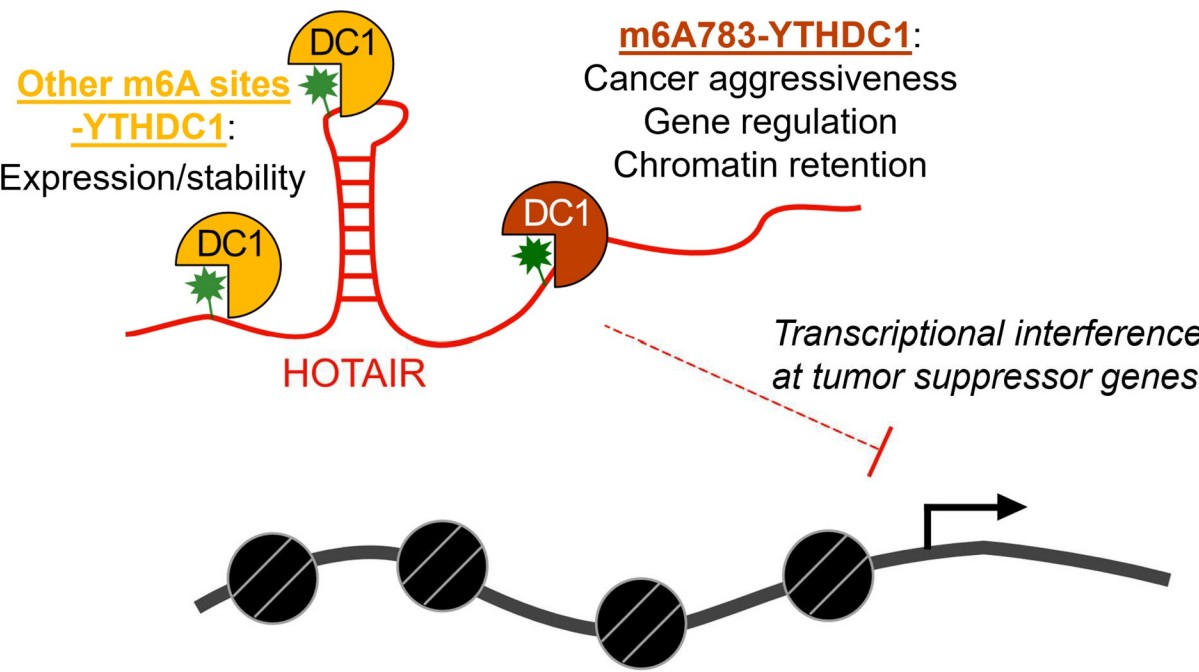

**Fig 7. Model of m6A and YTHDC1 effects on HOTAIR.** The main function of YTHDC1 on HOTAIR occurs via interaction with m6A783 and mediates chromatin association of HOTAIR to induce transcriptional interference of its target genes, promoting breast cancer growth and invasion. YTHDC1 also interacts with other m6A sites within HOTAIR that mediates its high expression levels and/or stability.

or the RNA molecules it binds to, dictate the effects of YTHDC1 association with its targets. Additional studies on how YTHDC1 interacts with specific RNA targets, chromatin, and other proteins in the nucleus will shed light on the mechanisms of YTHDC1 in chromatin regulation.

### Antimorphic transformation of HOTAIR function via mutation of a single m6A site

The antimorphic effect of mutating A783 in HOTAIR induced opposite and additional gene expression changes that ultimately resulted in a less aggressive breast cancer state (Figs 1 and 5). Our results show that disruption of a single m6A site can convert HOTAIR from eliciting pro- to antitumor effects, allowing overexpression of unmethylated lncRNA to decrease cancer phenotypes more so than depletion of the WT version. Understanding the mechanism behind this induction of antimorphic behavior by a single nucleotide mutation and its biological implications will require future work. One interesting observation is that the antimorphic lncRNA has lower overall retention on chromatin. This may potentiate the RNA to act at additional sites in the genome, with overall reduced occupancy at any one locus, explaining how more genes are affected with A783U than WT HOTAIR. Altogether, these findings suggest a potential approach to prevent the action of oncogenic lncRNAs such as HOTAIR, where disruption of RNA methylation alone has a greater impact than simple elimination of the RNA.

## Conclusion

The context dependency of m6A function is an emerging theme. With various roles in pluripotency and development and in disease states such as cancer, m6A on different RNA molecules regulates their fate in different ways [55,56]. Our work illustrates the context of 3 specific m6A functions on HOTAIR: enabling chromatin association, promoting high levels of lncRNA expression, and facilitating transcriptional repression. We further highlight the importance of one specific m6A site within an lncRNA that contains multiple sites of modification. This specific site, bound by YTHDC1, is critical for promoting HOTAIR-dependent cancer phenotypes and gene expression in TNBC cells. Overall, our work provides insight into mechanisms of how m6A regulates HOTAIR-mediated breast cancer metastasis which could ultimately lead to new strategies (for example, preventing m6A methylation at this specific site) to prevent or reverse the effects of elevated HOTAIR in breast cancer.

## Materials and methods

### Cell culture

MCF-7 cells were maintained in RPMI media (11875093, Invitrogen) and MDA-MB-231 and 293T in DMEM media (MT10013CV, Thermo Fisher Scientific). UCD cell lines were maintained in DMEM/F12 media (0-091-CV, Corning) supplemented with 100 ng/ml choleratoxin (BML-G117-0001, Enzo Life Sciences) and 1 nM insulin (12585014, Gibco). Media contained 10% FBS (F2442-500ML, Sigma-Aldrich) and 5% Pen-Strep (MT30002CI, Thermo Fisher Scientific) and cells were grown under standard tissue culture conditions. Cells were split using Trypsin (MT25053CI, Thermo Fisher Scientific) according to the manufacturer's instructions.

MDA-MB-231 cells overexpressing WT HOTAIR, A783U mutant HOTAIR, or anti-luciferase were generated as previously described using retroviral transduction [9]. Stable knockdown of METTL3, METTL14, WTAP, and YTHDC1 and overexpression of YTHDC1 was performed by lentivirus infection of MCF-7 cells or MDA-MB-231 cells overexpressing HOTAIR or A783U mutant HOTAIR via Fugene HD R.8 with pLKO.1-blasticidin shRNA

constructs or a pLX304 overexpression construct as noted in S4 Table. Cells were selected with 5 μg/mL blasticidin (Life Technologies). The non-targeting shRNA pLKO.1-blast-SCRAMBLE was obtained from Addgene (Catalog #26701). Two shRNAs for each target were obtained and stable lentiviral transductions with the targeted shRNAs and the scramble control were performed. Cell lines with the most efficient knockdown as determined by western blot with antibodies for METTL3 (15073-1-AP, Proteintech), METTL14 (HPA038002, Sigma), or WTAP (10200-1-AP, Proteintech) were selected for downstream experiments.

## Plasmid construction

The pBABE-puro retroviral vector was used for overexpression of lncRNAs. The spliced HOTAIR transcript (NR_003716.3) was synthesized and cloned into the pBABE-puro retroviral vector by GenScript. An antisense transcript of the firefly luciferase gene (AntiLuc) was amplified from the pTRE3G-Luciferase plasmid (Clonetech), and then cloned into the pBABE-puro retroviral vector. These were generated in a previous publication [9].

To create the A783U mutant HOTAIR overexpression plasmid, staggered QuikChange oligos AG66/AG67 were used to generate the A783U mutation in pTRE3G-HOTAIR using the QuikChange Site Directed Mutagenesis Kit (Agilent 200519) to generate pTRE3G-A783U_-HOTAIR. A 1.6 Kb fragment of A783U mutant HOTAIR was amplified with primers AG68/AG69 from pTRE3G-A783U_HOTAIR for cloning into pBABE-Puro-HOTAIR cut with XcmI and BamHI by Gibson Assembly. Oligonucleotide sequences are noted in S5 Table. All constructs were confirmed by sequencing. pBABE-Puro-6xAU_HOTAIR and pBabe-Puro-14xAU_HOTAIR were synthesized and cloned by GenScript.

Plasmids for the knockdown of METTL3, METTL14, WTAP, and YTHDC1 were generated by cloning the shRNA (RNAi Consortium shRNA Library) from pLKO.1-puro into the pLKO.1-blast backbone (Addgene #26655).

To generate the plasmid for tethering YTHDC1 to HOTAIR via dCasRX, we first constructed a pCDNA-FLAG plasmid by inserting a 5xFLAG sequence (synthesized as a gBlock by IDT DNA) into the HindIII/XbaI site of pCDNA3 (Invitrogen). YTHDC1 was then amplified from pLX304-YTHDC1 (ORF clone ccsdBroad304_04559) with oligonucleotides noted in S5 Table, and cloned into the KpnI/NotI site of pCDNA-FLAG to generate pCDNA-FLAG-YTHDC1 (pAJ367). The FLAG-YTHDC1 sequence was amplified then cloned downstream of dCasRX at NheI in the pXR002 plasmid (pXR002: EF1a-dCasRx-2A-EGFP was a gift from Patrick Hsu (Addgene plasmid # 109050; http://n2t.net/addgene:109050; RRID: Addgene_109050)) using oligonucleotides noted in S5 Table. Expression of the dCasRX-YTHDC1 fusion protein was confirmed by transfection of the plasmid followed by western blot with anti-FLAG M2 mouse monoclonal antibody (F1804, Sigma-Aldrich) and anti-YTHDC1 (14392-1-AP, Proteintech). Plasmids containing guide RNAs were generated using the pXR003 backbone plasmid (pXR003: CasRx gRNA cloning backbone was a gift from Patrick Hsu (Addgene plasmid # 109053; http://n2t.net/addgene:109053; RRID: Addgene_109053)) cut with BbsI, using oligonucleotides noted in S4 Table. All plasmids were confirmed by sequencing.

## m6A enhanced crosslinking immunoprecipitation

**PolyA isolation and RNA fragmentation.** For each experiment, approximately 100 μg of total RNA was isolated from cells with TRIzol according to the manufacturer's instructions. Approximately 10 μg PolyA RNA was isolated using Magnosphere Ultrapure mRNA Purification Kit (Takara) according to the manufacturer's instructions. PolyA RNA was ethanol precipitated with 2.5 M ammonium acetate and 70% ethanol in a solution containing 50 μg/ml

GlycoBlue Co-precipitant (AM9515, Invitrogen). RNA was resuspended in 10 μl and fragmented with 10× fragmentation buffer (AM8740, Invitrogen) at 75°C for 8 minutes and immediately quenched with 10× Stop Reagent (AM8740, Invitrogen) and placed on ice to generate fragments 30 to 150 nucleotides in length.

**Anti-m6A-RNA crosslinking and bead conjugation.** Crosslinked RNA-Antibody was generated as previously described [57]. Fragmented RNA was resuspended in 500 μl binding/ low salt buffer (50 mM Tris-HCl (pH 7.4), 150 mM sodium chloride, 0.5% NP-40) containing 2 μl Rnase Inhibitor (M0314, NEB) and 10 μl m6A antibody (ab151230, Abcam), and incubated for 2 hours at room temperature with rotation. RNA-Antibody sample was transferred to 1 well of a 12-well dish and placed in a shallow dish of ice. Sample was crosslinked twice at 150 mJ/cm$^2$ using a Stratagene Stratalinker UV Crosslinker 1800 and transferred to a new tube. A total of 50 μl Protein A/G Magnetic Beads (88803, Pierce) were washed twice with binding/low salt buffer, resuspended in 100 μl binding/low salt buffer, and added to crosslinked RNA-Antibody sample. Beads were incubated at 4°C overnight with rotation.

**eCLIP library preparation.** RNA was isolated and sequencing libraries were prepared using a modified enhanced CLIP protocol [58]. Beads were washed twice with high salt wash buffer (50 mM Tris-HCl (pH 7.4), 1 M sodium chloride, 1 mM EDTA, 1% NP-40, 0.5% sodium deoxycholate, 0.1% sodium dodecyl sulfate), once with wash buffer (20 mM Tris-HCl (pH 7.4), 10 mM magnesium chloride, 0.2% Tween-20), once with wash buffer and 1× fast AP buffer (10 mM Tris (pH 7.5), 5 mM magnesium chloride, 100 mM potassium chloride, 0.02% Triton X-100) combined in equal volumes, and once with 1× fast AP buffer. Beads were resuspended in Fast AP Master Mix (1× fast AP buffer containing 80U Rnase Inhibitor (M0314, NEB), 2U TURBO Dnase (AM2238, Invitrogen), and 8U Fast AP Enzyme (EF0654, Thermo Fisher Scientific)) was added. Samples were incubated at 37°C for 15 minutes shaking at 1,200 rpm. PNK Master Mix (1× PNK buffer (70 mM Tris-HCl (pH 6.5), 10 mM magnesium chloride), 1 mM dithiothreitol, 200U Rnase Inhibitor, 2U TURBO Dnase, 70U T4 PNK (EK0031, Thermo Fisher Scientific)) was added to the samples and they incubated at 37°C for 20 minutes shaking at 1,200 rpm.

Beads were washed once with wash buffer, twice with wash buffer and high salt wash buffer mixed in equal volumes, once with wash buffer, once with wash buffer and 1× ligase buffer (50 mM Tris (pH 7.5), 10 mM magnesium chloride) mixed in equal volumes, and twice with 1× ligase buffer. Beads were resuspended in Ligase Master Mix (1× ligase buffer, 1 mM ATP, 3.2% DMSO, 18% PEG 8000, 16U Rnase Inhibitor, 75U T4 RNA Ligase I (M0437, NEB)), 2 barcoded adaptors were added (X1a and X1b, see S6 Table), and samples were incubated at room temperature for 75 minutes with flicking every 10 minutes. Beads were washed once with wash buffer, once with equal volumes of wash buffer and high salt wash buffer, once with high salt wash buffer, once with equal volumes of high salt wash buffer and wash buffer, and once with wash buffer. Beads were resuspended in wash buffer containing 1× NuPAGE LDS sample buffer (NP0007, Invitrogen) and 0.1 M DTT, and incubated at 70°C for 10 minutes shaking at 1,200 rpm.

Samples were cooled to room temperature and supernatant was ran on Novex NuPAGE 4% to 12% Bis-Tris Gel (NP0321, Invitrogen). Samples were transferred to nitrocellulose membrane, and membranes were cut and sliced into small pieces between 20 kDa and 175 kDa to isolate RNA-antibody complexes. Membrane slices were incubated in 20% Proteinase K (03508838103, Roche) in PK buffer (100 mM Tris-HCl (pH 7.4), 50 mM NaCl, 10 mM EDTA) at 37°C for 20 minutes shaking at 1,200 rpm. PK buffer containing 7 M urea was added to samples and samples were incubated at 37°C for 20 minutes shaking at 1,200 rpm. Phenol:Chloroform:Isoamyl Alcohol (25:24:1) (P2069, Sigma-Aldrich) was added to samples, and samples

were incubated at 37˚C for 5 minutes shaking at 1,100 rpm. Samples were centrifuged 3 minutes at 16,000 × $g$ and aqueous layer was transferred to a new tube.

RNA was isolated using RNA Clean and Concentrator-5 Kit (R1016, Zymo) according to the manufacturer's instructions. Reverse transcription was performed using AR17 primer (S5 Table) and SuperScript IV Reverse Transcriptase (18090010, Invitrogen). cDNA was treated with ExoSAP-IT Reagent (78201, Applied Biosystems) at 37˚C for 15 minutes, followed by incubation with 20 mM EDTA and 0.1 M sodium hydroxide at 70˚C for 12 minutes. Reaction was quenched with 0.1 M hydrochloric acid. cDNA was isolated using Dynabeads MyONE Silane (37002D, Thermo Fisher Scientific) according to the manufacturer's instructions. Approximately 20% DMSO and rand3Tr3 adaptor (S5 Table) was added to samples, and samples were incubated at 75˚ for 2 minutes. Samples were placed on ice and Ligation Master Mix (1× NEB ligase buffer, 1 mM ATP, 25% PEG 8000, 15U T4 RNA Ligase I (NEB)) was added to samples. Samples were mixed at 1,200 rpm for 30 seconds prior to incubation at room temperature overnight.

cDNA was isolated using Dynabeads MyONE Silane according to the manufacturer's instructions and eluted with 10 mM Tris-HCl (pH 7.5). A 1:10 dilution of cDNA was used to quantify the cDNA library by qPCR using a set of Illumina's HT Seq primers, and Ct values were used to determine number of cycles for PCR amplification of cDNA. The undiluted cDNA library was amplified by combining 12.5 μl of the sample with 25 μl Q5 Hot Start PCR Master Mix and 2.5 μl (20 μm) of the same indexed primers used previously. Amplification for the full undiluted sample used 3 cycles less than the cycle selected from the diluted sample. The PCR reaction was isolated using HighPrep PCR Clean-up System (AC-60050, MAGBIO) according to the manufacturer's instructions.

The final sequencing library was gel purified by diluting the sample with 1× Orange G DNA loading buffer and running on a 3% quick dissolve agarose gel containing SYBR Safe Dye (1:10,000). Following gel electrophoresis, a long wave UV lamp was used to extract DNA fragments from the gel ranging from 175 to 300 base pairs. The DNA was isolated using QiaQuick MinElute Gel Extraction Kit (28604, Qiagen). The purified sequencing library was analyzed via TapeStation using DNA ScreenTape (either D1000 or HS D1000) according to the manufacturer's instructions to assess for appropriate size and concentration (the final library should be between 175 and 300 base pairs with an ideal concentration of at least 10 nM).

**Sequencing and analysis.** Samples were sequenced at the Genomics and Microarray Shared Resource facility at University of Colorado Denver Cancer Center on an Illumina MiSeq or NovaSEQ6000 with 2× 150 base pair paired-end reads to generate 40 million raw reads for each sample. Computational analysis methods are described in [31]. Briefly, a custom Snakemake workflow was generated based on the original eCLIP analysis strategies [58] to map reads to the human genome. To identify m6A sites, we used a custom analysis pipeline to identify variations from the reference genome at single-nucleotide resolution across the entire genome. We then employed an internally developed Java package to identify C-to-T mutations occurring (1) within the m6A consensus motif "RAC": "R" is any purine, A or G; A being the methylated adenosine; and C where the mutation occurs; and (2) within a frequency range of greater than or equal to 2.5% and less than or equal to 50% of the total reads at a given position (with a minimum of 3 C-to-T mutations at a single site). The resulting m6A sites were then compared to those identified in the corresponding input sample and any sites occurring in both were removed from the final list of m6A sites (this eliminates any mutations that are not directly induced from the anti-m6A antibody crosslinking). Full transcriptome data associated with the methods manuscript [31] is at GEO accession number GSE147440.

## m6A RNA immunoprecipitation (meRIP)

Total RNA was isolated with TRIzol (15596018, Invitrogen) according to the manufacturer's instructions. RNA was diluted to 1 µg/µl and fragmented with 1× fragmentation buffer (AM8740, Invitrogen) at 75°C for 5 minutes; 1× Stop Reagent (AM8740, Invitrogen) was added immediately following fragmentation and samples placed on ice. Approximately 500 ng of input sample was reserved in 10 µl nuclease free water for qRT-PCR normalization. Protein A/G Magnetic Beads (88803, Pierce) were washed twice with IP buffer (20 mM Tris (pH 7.5), 140 mM NaCl, 1% NP-40, 2 mM EDTA) and coupled with anti-m6A antibody (ab151230, Abcam) or an IgG control (NB810-56910, Novus) for 1 hour at room temperature. Beads were washed 3 times with IP buffer. Approximately 10 µg fragmented RNA and 400U Rnase inhibitor was added to 1 ml IP buffer. Antibody-coupled beads were resuspended in 500 µl RNA mixture and incubated 2 hours to overnight at 4°C on a rotor. Beads were washed 5 times with cold IP buffer. Elution buffer (1× IP buffer containing 10 U/µl Rnase inhibitor and 0.5 mg/ml N6-methyladenosine 5′-monophosphate (M2780, Sigma-Aldrich)) was prepared fresh and kept on ice. Samples were eluted with 200 µl elution buffer for 2 hours at 4°C on a rotor. Supernatant was removed and ethanol precipitated with 2.5 M ammonium acetate, 70% ethanol, and 50 µg/ml GlycoBlue Coprecipitant (Invitrogen AM9515). RNA was washed with 70% ethanol, dried for 10 minutes at room temperature, and resuspended in 10 µl nuclease free water. RNA was quantified by nanodrop and 200 ng RNA was reverse transcribed using High Capacity cDNA Reverse Transcription Kit (4368814, Thermo Fisher Scientific) and quantified by qPCR (oligonucleotides listed in S7 Table), and fraction recovered was calculated from Input and IP values.

## RNA immunoprecipitation of YTHDC1

Actively growing cells from 70% to 90% confluent 15-cm dishes were trypsinized and washed twice with ice-cold 1× PBS. Cell pellet was resuspended in 1% V/V formaldehyde (28908, Pierce) in 1× PBS and incubated at room temperature for 10 minutes on a rotor. Crosslinking was quenched with 0.25 M glycine at room temperature for 5 minutes. Cells were washed 3 times with ice-cold 1× PBS and placed on ice. Approximately 20 µl Protein A/G beads were washed twice with RIPA binding buffer (50 mM Tris-HCl (pH 7.4), 100 mM sodium chloride, 1% NP-40, 0.1% sodium dodecyl sulfate, 0.5% sodium deoxycholate, 4 mM dithiothreitol, 1× protease inhibitors), resuspended in 1 ml RIPA binding buffer, and split to two 0.5 ml aliquots. Approximately 2 µg YTHDC1 antibody (ab122340, Abcam) or an IgG Control (sc-2027, Santa Cruz Biotechnology) was added to beads and incubated for 2 hours at 4°C on a rotor. Fixed cells were resuspended in 1 ml RIPA binding buffer and placed in the Bioruptor Pico (B01060010, Diagenode) for 10 cycles of 30 seconds on, 30 seconds off. Lysates were digested with TURBO Dnase for 5 minutes at 37°C with mixing at 1,000 rpm and transferred to ice for 5 minutes. Lysates were clarified by centrifugation at 17,000$g$ at 4°C for 10 minutes and supernatant was transferred to a new tube. The 200U Rnase Inhibitor was added to the 1 ml clarified lysate. A 5% aliquot was removed and processed downstream with IP samples. A 2% aliquot was removed and diluted with 1× SDS sample buffer (62.5 mM Tris-HCl (pH 6.8), 2.5% SDS, 0.002% Bromophenol Blue, 5% β-mercaptoethanol, 10% glycerol) and protein input and recovery was monitored by western blot. Antibody-coupled beads were washed 3 times with RIPA binding buffer and resuspended in half of the remaining lysate. Samples were incubated overnight at 4°C on a rotor. Beads were washed 5 times with RIPA wash buffer (50 mM Tris-HCl (pH 7.4), 1 M sodium chloride, 1% NP-40, 0.1% sodium dodecyl sulfate, 0.5% sodium deoxycholate, 1 M Urea, 1× protease inhibitors) and resuspended in 100 µl RNA elution buffer (50 mM Tris-HCl (pH 7.4), 5 mM EDTA, 10 mM dithiothreitol, 1% sodium dodecyl sulfate).

Input sample was diluted with 1× RNA elution buffer. Formaldehyde crosslinks in both input and IP samples were reversed by incubation at 70°C for 30 minutes at 1,000 rpm. Supernatant was transferred to a new tube and RNA was isolated using TRIzol-LS according to the manufacturer's instructions. Reverse transcription was performed on 100 ng RNA using SuperScript IV Reverse Transcriptase. qPCR was performed as described below.

## RNA isolation and qRT-PCR

RNA was isolated with TRIzol (Life Technologies) with extraction in chloroform followed by purification with the Rneasy kit (Qiagen). Samples were Dnase treated using TURBO Dnase (Ambion). Reverse transcription was performed using the cDNA High Capacity Kit (Life Technologies). qPCR was performed using Sybr Green master mix (Takyon, AnaSpec) using the primers listed in S6 Table on a C1000 Touch Thermocycler (BioRad). The HOTAIR qPCR primer set targeting region 499–668 obtained from Portoso and colleagues [15] was used to quantify HOTAIR except where otherwise noted. EEF1A1 primer sequences were obtained from the Magna MeRIP m6A kit (17–10499, Sigma-Aldrich). Sequences for Luciferase primers (LucR2) were obtained from a previous publication [59]. Three qPCR replicates were performed for each sample, and these technical replicates were averaged prior to analysis of biological replicates. At least 3 biological replicates were performed for each qPCR experiment.

## Cell proliferation assays

At least 2 independent clones of each transgenic cell line, here defined as a pool of selected cells stably expressing the pBabe plasmid, were analyzed for cell proliferation. A total of 2,000 cells were plated in a 96-well dish in DMEM media containing 10% FBS and selective antibiotics (1 μg/ml puromycin (P8833, Sigma-Aldrich) and/or 5 μg/ml blasticidin (71002–676, VWR)), allowed to settle at room temperature for 20 minutes, then placed in an Incucyte S3 (Sartorius). Pictures were taken with a 10× magnification every 2 hours for 48 hours using a standard scan. Confluency was determined using the Incucyte ZOOM software. Growth rate was calculated from % confluency using the least squares fitting method [60].

## Cell invasion assays

MDA-MB-231 cell lines were grown to 70% to 90% confluence and serum starved in Opti-MEM for approximately 20 hours prior to setting up the experiment. Cells were washed, trypsinized, and resuspended in 0.5% serum DMEM. Approximately 10% serum DMEM was added to the bottom chamber of Corning Matrigel Invasion Chambers (Corning 354481), and 40,00 cells for pBabe transgenic lines, or 100,000 to 300,000 cells for dCasRX transfection experiments, were plated in the top chamber in 0.5% serum DMEM. Cells were incubated for 22 hours at 37°C followed by 4% PFA fixation and 0.1% crystal violet staining. Matrigel inserts were allowed to dry overnight, followed by brightfield imaging with a 20× air objective. Four biological replicates were performed, with technical duplicates in each set. For each Matrigel insert, 4 fields of view were captured, and cells were counted in Fiji (8 data points per condition, per biological replicate). The violin plots include all of the data points, while statistical analysis was performed on the average number of cells/field for each biological replicate.

## Purification of METTL3/14

Suspension-adapted HEK293 cells (Freestyle 293-F cells, R790-07, Life Technologies) were grown as recommended in Freestyle 293 Expression Medium (12338026, Life Technologies,) shaking at 37°C in 5% $CO_2$. Cells were grown to a concentration of $3 \times 10^6$ cells/ml and diluted

to $1 \times 10^6$ cells/ml in 50 ml 293F Freestyle Media 24 hours prior to transfection. Before transfection, cells were spun down and resuspended in 50 ml fresh 293F Freestyle Media at a concentration of $2.5 \times 10^6$ cells/ml. Expression plasmids (pcDNA3.1-FLAG-METTL3, pcDNA3.1-FLAG-METTL14) were added to the flask at a concentration of 1.5 μg/ml, and flask was shaken in the incubator for 5 minutes. Approximately 9 μg/ml PEI was added to the flask and cells were returned to incubator. After 24 hours of growth, an additional 50 ml fresh 293F Freestyle Media was added and culture was supplemented with 2.2 mM VPA. Cells were harvested as two 50 ml pellets 72 hours after addition of VPA.

Cell pellets were resuspended in 1× lysis buffer (50 mM Tris (pH 7.4), 150 mM sodium chloride, 1 mM EDTA, 1% TritonX-100, 1× protease inhibitors) to obtain a concentration of $10^7$ cells/ml and incubated for 20 minutes at 4˚C with rotation. Cell lysate was clarified by centrifugation at 4˚C, $12,000 \times g$ for 15 minutes. Supernatant was transferred to a new tube and kept on ice. Anti-FLAG M2 affinity resin (A2220, Sigma-Aldrich) was equilibrated with 1× lysis buffer by washing 3 times. Equilibrated resin was resuspended in 1× lysis buffer and added to the tube containing the clarified lysate. Sample was incubated for 2 hours at 4˚C with rotation. Resin was pelleted by centrifugation at 4˚C, $500 \times g$. Supernatant was removed, and resin was washed 3 times with 1× wash buffer (50 mM Tris (pH 7.4), 150 mM sodium chloride, 10% glycerol, 1 mM dithiothreitol) for 5 minutes each at 4˚C with rotation. Sample was equilibrated to room temperature, and resin was resuspended in 1× wash buffer containing 0.2 mg/ml 3xFLAG peptide. Samples were incubated at room temperature for 10 minutes shaking at 1,000 rpm, centrifuged for 2 minutes at $1000 \times g$, and supernatant was reserved (elution 1). Elution was repeated twice to obtain 2 additional elution samples (elution 2 and 3). Samples were analyzed by Coomassie to determine protein concentration and purity. Samples were aliquoted and stored at −80˚C and thawed on ice prior to use in in vitro m6A methylation experiments.

## In vitro m6A methylation and interaction assays

All plasmids and oligonucleotides used in this assay are listed in S8 Table. Using PCR, we generated a DNA fragment for Domain 2 of WT (pTRE3G-HOTAIR, pAJ171) and A783U (pTRE3G-A783U_HOTAIR, pAJ385) mutant HOTAIR using primers MB88 and MB89. A 5′ T7 promoter and 3′ RAT tag were added to the sequence via PCR with primers MB22 and MB94. In vitro transcription of the PCR templates was completed using the MEGAScript T7 Transcription Kit (AM1334, Thermo Fisher Scientific) according to the manufacturer's instructions, RNA was purified using the Rneasy Mini Kit (Qiagen 75106) and quantified by UV. Approximately 500 nM RNA was diluted in 1× methyltransferase buffer (20 mM Tris (pH 7.5), 0.01% Triton-X 100, 1 mM DTT) in reactions containing 50 μm SAM and 500 nM purified METTL3/14 (+m6A) for 1 hour at room temperature. Control reactions contained no METTL3/14 (-m6A). RNA was purified using the Rneasy Mini Kit according to the manufacturer's instructions and quantified by UV.

To obtain FLAG-tagged YTHDC1 protein, 293 cells were transfected using Lipofectamine 2000 (11668030, Thermo Fisher Scientific) with plasmid pAJ367 pCDNA-FLAG-YTHDC1 and cell lysates were generated as previously described [9]. Dynabeads (M270, Invitrogen) were resuspended in high-quality dry dimethylformamide at a concentration of $2 \times 10^9$ beads/ml. Dynabeads were stored at 4˚C and equilibrated to room temperature prior to use. Dynabeads were washed in 0.1 M sodium phosphate buffer (pH 7.4) and vortexed for 30 seconds. A second wash was repeated with vortexing and incubation at room temperature for 10 minutes with rotation. Approximately 1 mg/ml IgG solution was prepared by diluting rabbit IgG (15006, Sigma) in 0.1 M sodium phosphate buffer. Washed beads were resuspended in 0.1 M

sodium phosphate buffer at a concentration of $3 \times 10^9$ beads/ml, and an equal volume of 1 mg/ml IgG was added. Samples were vortexed briefly and an equal volume of 3M ammonium sulfate was added and samples were mixed well. Samples were incubated at 37˚C for 18 to 24 hours with rotation. Samples were washed once briefly with 0.1 M sodium phosphate buffer, then twice with incubation at room temperature for 10 minutes with rotation. Samples were washed in sodium phosphate buffer + 1% TritonX-100 at 37˚C for 10 minutes with rotation. A quick wash with 0.1 M sodium phosphate buffer was performed and followed by 4 washes in 0.1 M citric acid (pH 3.1) at a concentration of $2 \times 10^8$ beads/ml at room temperature for 10 minutes with rotation. After a quick wash with 0.1 M sodium phosphate buffer, beads were resuspended to $1 \times 10^9$ beads/ml in 1× PBS + 0.02% sodium azide and stored at 4˚C prior to use.

Approximately 800 ng of +/-m6A RNA was incubated with 150 ng PrA-PP7 fusion protein in HLB300 (20 mM Hepes (pH 7.9), 300 mM sodium chloride, 2 mM magnesium chloride, 0.1% NP-40, 10% glycerol, 0.1 mM PMSF, 0.5 mM DTT). RNA was prebound to PP7 for 30 minutes at 25˚C, 1,350 rpm. A total of 75 µl IgG-coupled Dynabeads were washed with HLB300 twice and resuspended in 250 µl HLB300. Approximately 50 µl beads were added to each tube of RNA-PP7 and samples were incubated 1 hour at 25˚C, 1,350 rpm. Beads were washed twice with HLB300 and resuspended in 80 µl binding buffer (10 mM Hepes (pH 7.4), 150 mM potassium chloride, 3 mM magnesium chloride, 2 mM DTT, 0.5% NP-40, 10% glycerol, 1 mM PMSF, 1× protease inhibitors) containing 80U Rnase Inhibitor. A total of 25 µg YTHDC1-FLAG containing lysate and 800 ng competitor RNA (IVT untagged HOTAIR D2) was added to each sample. Samples were incubated at 4˚C for 2.5 hours on a rotor. Beads were washed 3 times with cold wash buffer (200 mM Tris-HCl (pH 7.4), 200 mM sodium chloride, 2 mM magnesium chloride, 1 mM DTT, 1× protease inhibitors) and resuspended in 1× SDS loading buffer. A 10% protein input sample was diluted in 1× SDS loading buffer. Samples were boiled 5 minutes at 95˚C and supernatant transferred to a new tube. Half of each sample was loaded on a 10% acrylamide gel and western blot was performed using anti-FLAG M2 mouse monoclonal antibody (F1804, Sigma-Aldrich).

## Fractionation

Cells were grown in 15-cm dishes to 70% to 90% confluency. Cells were released with Trypsin (Corning), quenched with media, washed once with 1× PBS containing 1 mM EDTA, and split into 2 volumes; ¼ of the sample was harvested in TRIzol and RNA isolated with Rneasy kit for the input RNA sample. The remaining ¾ of the sample was fractionated into cytoplasmic, nucleoplasmic, and chromatin-associated samples. Cells were lysed in cold cell lysis buffer (10 mM Tris-HCl (pH 7.5), 0.15% NP-40, 150 mM sodium chloride) containing Rnase inhibitors for 5 minutes on ice. Lysate was layered onto 2.5 volumes of sucrose cushion (10 mM Tris-HCl (pH 7.5), 150 mM sodium chloride, 24% sucrose) containing Rnase inhibitors. Samples were centrifuged for 10 minutes at $17,000 \times g$ at 4˚C. Supernatant was collected (cytoplasmic sample). Pellet was rinsed with 1× PBS containing 1 mM EDTA and resuspended in cold glycerol buffer (20 mM Tris-HCl (pH 7.9), 75 mM sodium chloride, 0.5 mM EDTA, 0.85 mM DTT, 0.125 mM PMSF, 50% glycerol) containing Rnase inhibitors. An equal volume of cold nuclei lysis buffer (10 mM HEPES (pH 7.6), 1 mM DTT, 7.5 mM magnesium chloride, 0.2 mM EDTA, 0.3 M sodium chloride, 1 M Urea, 1% NP-40) was added and sample was briefly vortexed twice for 2 seconds. Samples were incubated on ice 2 minutes and centrifuged for 2 minutes at $17,000 \times g$ at 4˚C. Supernatant was collected (nucleoplasmic sample). The remaining pellet was resuspended in 1× PBS containing 1 mM EDTA (chromatin-associated sample). Each sample was subjected to TURBO Dnase digestion at 37˚C for 30 minutes in 1× TURBO

buffer and 10U TURBO for cytoplasmic and nucleoplasmic samples, or 40U TURBO for chromatin-associated sample. Reactions were quenched with 10 mM EDTA, fractions removed for analysis by western blot, and 3 volumes of TRIzol-LS were added. RNA isolation was performed as recommended by the manufacturer. Samples were quantified by nanodrop to determine RNA concentration and ran on a 2% agarose gel to confirm RNA integrity. qRT-PCR was performed on 2 μg of RNA and normalized to RNA recovery, input values, and GAPDH. Western blot was performed on fractions to analyze fractionation efficiency with antibodies against cytoplasmic B-actin (66009–1, Proteintech), nuclear hnRNP A2B1 (NB120-6102, Novus), and chromatin-associated H3 (ab21054, Abcam).

## Luciferase assay

Analysis of luciferase activity was performed using the Luciferase Assay System (E1500, Promega). Cells were washed with 1× PBS and lysed in 100 μl 1× Cell Culture Lysis Reagent. Cells were scraped from bottom of dish and suspension was transferred to a new tube. Lysates were frozen and thawed prior to luciferase assay to ensure complete lysis. Luciferase assays were performed on 20 μl of lysate or 1× Cell Culture Lysis Reagent in 96-well plates on the GloMax-Multi Detection System (TM297, Promega). A total of 100 μl Luciferase Assay Reagent was added to wells, mixed, and light production measured. Measurements were performed in 3 technical replicates for each biological replicate. Luciferase activity was normalized to protein concentration of samples.

## siRNA transfection

Silencer Select siRNAs were obtained from Thermo Fisher Scientific targeting YTHDC1 (n372360, n372361, n372362) or Negative Controls (4390843, 4390846) and transfected into 293 cell lines using Lipofectamine RNAiMAX Transfection Reagent (13778030, Thermo Fisher Scientific). Transfections were performed in a 24-well plate with 5 pmol of siRNA and 1.5 μl RNAiMAX Transfection reagent per well. Cells were harvested 24 hours after transfections and analyzed by Luciferase Assay and qRT-PCR.

## Gene expression analyses

Total RNA was extracted from MDA-MB-231 cells using TRIZol (Life Technologies) with extraction in chloroform followed by purification with the RNeasy kit (Qiagen). Samples were DNase treated using TURBO DNase (Ambion). PolyA-selected sequencing libraries were prepared and sequenced by The Genomics Shared Resource at the University of Colorado Cancer Center. All gene expression data associated with this publication are available through GEO accession number GSE173530. Differential gene expression analysis was performed using Salmon and DESeq2 [61,62]. Briefly, the reads were quantified using Salmon to generate transcript abundance estimates and then DESeq2 was used to determine differential expression between samples. Heat maps were generated by using normalized read counts of genes that were significantly ($p < 0.1$) differentially expressed between conditions to generate Z-scores. GO term enrichment analysis was performed using the GO Consortium's online PANTHER tool [63–65]. To analyze correlation between expression in HOTAIR v. Control and A783U v. Control pairwise comparisons, the total set of DEGs were filtered to include only those whose fold change value was greater than 1.15 in either direction for both comparisons. These values were then plotted against each other. Linear regression was used to fit a trend line over the points, with the calculated Pearson correlation coefficient included in the graph.

## dCasRX-YTHDC1 and gRNA transfection

One plasmid containing dCasRX-FLAG-YTHDC1 in pXR002 in combination with 1 plasmid containing the designated guide RNA in pXR003 (see description in Plasmid Construction) were transfected into a 70% to 90% confluent 10-cm dish using Lipofectamine 2000 (11668030, Invitrogen) according to the manufacturer's instructions. Plates were incubated at 37°C for approximately 24 hours, and then subjected to fractionation as described above.

## Statistical analyses

Numbers in text report mean ± standard deviation for all data. Graphs were prepared and data fitting and statistical analyses were performed using Biovinci (version 1.1.5, Bioturing, San Diego, California, USA). Each box-and-whisker plot displays data points for each replicate, the median value as a line, a box around the lower and upper quartiles, and whiskers extending to maximum and minimum values, excluding outliers as determined by the upper and lower fences. Violin plots are similar to box plots, except that they also show the probability density of the data at different values, smoothed by a kernel density estimator. In BioVinci, the Gaussian kernel is used as the estimator and Scott's rule is used to calculate the estimator bandwidth. A Student's 2-tailed unpaired $t$ test was used to determine the statistical significance between 2 samples. Differences and relationships were considered statistically significant when $p \leq 0.05$. For all graphs, $^{*}$ $p < 0.05$, $^{**}$ $p < 0.01$, $^{***}$ $p < 0.001$, $^{****}$ $p < 0.0001$.

## Supporting information

**S1 Data. Excel file of differentially expressed genes identified in DESeq2 pairwise comparisons and overlap of HOTAIR ChIRP peaks and HOTAIR-dependent H3K27me3 peaks with HOTAIR v. Control differentially expressed genes.**
(XLSX)

**S2 Data. Numerical values underlying summary data displayed in all main figure graphs including panels 1B–D, 1F–H, 2A–D, 2G, 3B–G, 4B–C, 4E–F, 4H–I, 5B–C, 5E, and 6C–G.**
(XLSX)

**S3 Data. Numerical values underlying summary data displayed in all supplemental figure graphs including panels S2A–S2E, S3B–S3D, S4D–S4F, S6D, S7A–S7B, and S8D.**
(XLSX)

**S1 Fig. Investigation of m6A in HOTAIR. (A)** CVm6A visualization of HOTAIR m6A RIP experiments. Data obtained from http://gb.whu.edu.cn:8080/CVm6A. **(B)** Schematic of m6A eCLIP pipeline used to map m6A sites. **(C)** Portion of HOTAIR structure from [16]. A783 is marked by a green arrow.
(TIF)

**S2 Fig. HOTAIR is m6A modified at A783 by METTL3/14 and regulates proliferation of TNBC cells. (A)** m6A RNA immunoprecipitation performed with an m6A antibody or IgG control in MCF-7 breast cancer cells, quantified with probe sets spanning regions of HOTAIR and EEF1A1 as noted. Four biological replicates were performed. **(B)** Relative HOTAIR levels in breast cancer cell lines as determined by qRT-PCR. **(C)** Left, western blot of knockdown lines generated in MCF-7 cells. Right, m6A RIP results on MCF-7 knockdown lines. **(D)** m6A RNA immunoprecipitation performed with an m6A antibody or IgG control on MDA-MB-231 cells overexpressing WT HOTAIR, or HOTAIR$^{A783U}$, quantified with probe sets spanning regions of HOTAIR and EEF1A1 as noted. Four biological replicates were performed. **(E)** Left, example of growth curve obtained from Incucyte experiments. Percent confluence was

measured every 2 hours for 48 hours on MDA-MB-231 cell lines noted. Right, data from Fig 1F, doubling time of MDA-MB-231 overexpression cell lines, displayed as individual data points (3 clones for each cell line noted, 3 biological replicates for each clone). Numerical values in panels S2A–S2E are included in S3 Data.
(TIF)

**S3 Fig. Mutations in HOTAIR that prevent m6A783 block HOTAIR's ability to promote proliferation and invasion of MDA-MB-231 cells. (A)** Schematic of mutations to block m6A modification of A783 in HOTAIR. **(B)** Relative HOTAIR levels in transgenic MDA-MB-231 cell lines overexpressing WT or mutant HOTAIR or an antisense-luciferase control, or MCF-7 cells, as noted. **(C)** Doubling time of MDA-MB-231 overexpression cell lines noted. Experiments include 2 biological replicates each on 2 independently generated clones. **(D)** Quantification of Matrigel invasion assays performed with MDA-MB-231 overexpression cell lines noted. Two biological replicates each on 2 independently generated clones were performed. **(E)** Analysis of experiments in B–D. Numerical values in panels S3B–S3D are included in S3 Data.
(TIF)

**S4 Fig. hnRNP B1 and YTHDF1/2 do not directly interact with HOTAIR m6A. (A)** Results of a search of 2,000 base pair regions surrounding m6A sites for hnRNP B1 eCLIP peaks using set functions. **(B)** hnRNP B1 eCLIP intensity relative to m6A sites that contain overlap with hnRNP B1 binding sites within 1,000 nucleotides upstream or downstream. **(C)** Map of HOTAIR qPCR probes, m6A sites, in vitro eCLIP peaks of hnRNP B1 binding, and hnRNP B1 eCLIP peaks in MCF-7 cells [13]. **(D)** YTHDC1 RIP performed in MDA-MB-231 cells overexpressing WT HOTAIR or HOTAIR$^{A783U}$, quantified with probe sets spanning regions of HOTAIR and EEF1A1 as noted. Recovery was normalized to region 1819–1923 where no m6A sites were detected. Four biological replicates were performed. **(E, F)** RIP performed in MCF-7 cells with antibodies against YTHDF1 (E) or YTHDF2 (F) on 3 biological replicates. RNA recovery was monitored with qPCR probes noted in graph and divided by recovery observed with qPCR probes targeting a region of HOTAIR with no m6A (1819–1923). Numerical values in panels S4D–S4F are included in S3 Data.
(TIF)

**S5 Fig. YTHDC1 expression levels in breast cancer regulate predictive nature of HOTAIR expression. (A, B)** Overall survival curves for breast cancer patients examining effect of HOTAIR on the background of either (E) high or (F) low median *YTHDC1* levels, generated with Kaplan–Meier plotter [39]. **(C, D)** Expression of YTHDC1 (A) mRNA and (B) protein in normal breast tissue versus breast cancers generated with UALCAN [40].
(TIF)

**S6 Fig. Fractionation of MDA-MB-231 cells, model for effects of YTHDC1 dosage in WT vs. A783U HOTAIR-expressing cells, and proliferation effects of multiple m6A mutants. (A)** Western blot on fractionation of MDA-MB-231 cell lines overexpressing WT or A783U HOTAIR or antisense-luciferase. **(B)** Western blot performed on fractionation of MDA-MB-231 cell lines overexpressing WT or A783U HOTAIR containing overexpression (OE), non-targeting (NT), or knock-down (KD) of YTHDC1. **(C)** Model for differences observed between WT and A783U HOTAIR upon knockdown and overexpression of YTHDC1. **(D)** Doubling time of MDA-MB-231 cells expressing WT, A783U, 6xAU, or 14xAU HOTAIR, or an anti-luciferase control. Three biological replicates each were performed on 2 independently generated clones. Numerical values in panels S6D are included in S3 Data.
(TIF)

**S7 Fig. HOTAIR is stable upon YTHDC1 knockdown in 293T HOTAIR-tethered reporter cell lines. (A, B)** qRT-PCR of HOTAIR (A) or MS2 (B) in 293T HOTAIR-tethered cells lacking EED with siRNA knockdown of METTL3 or YTHDC1 or a non-targeting control. **(C)** Model for HOTAIR stability in chromatin-tethered context. Numerical values in panels S7A–S7B are included in S3 Data.
(TIF)

**S8 Fig. Expression of A783U HOTAIR induces opposite gene expression changes compared to WT HOTAIR. (A)** Heat map of Z-scores of all DEGs identified in pairwise comparisons. **(B, C)** Venn diagrams (created using BioVenn [66]) of comparisons of DEGs by expression pattern noted. **(D)** Relative expression of HOTAIR gRNA samples compared to NT gRNA samples for WT HOTAIR-up-regulated genes noted. Numerical values in panel S8D are included in S3 Data.
(TIF)

**S1 Table. List of HOTAIR m6A sites by experiment.** Each column represents a single experiment with the cell line noted. Each row is an m6A site detected within HOTAIR and includes Nt # (location within HOTAIR transcript) and chromosome position. Each X represents an m6A site detected in HOTAIR in each experiment. Using thresholding from [31], "**X**" represents high confidence m6A sites ($\geq$3 C➔T mutations following the m6A site in $\geq$5% of reads), "x" represents low confidence sites ($\geq$3 C➔T mutations following the m6A site in $\geq$2.5% of reads), and "*" represents sites called with reduced threshold of at least 2 C➔T mutation events detected. HOTAIR m6A site positions in bold were included in the 6× HOTAIR mutant, while the remainder of sites (not including Nt 557) were included in the 14× HOTAIR mutant. Note that MCF-7 replicate 3 was a lower-depth run used as a point of comparison to the higher-depth replicates 1 and 2 in [31]. HOTAIR read depth in MCF-7 experiments is approximately 10× less than when overexpressed in MDA-MB-231 cells. m6A site 783 is highlighted in red.
(DOCX)

**S2 Table. Multiple replicate consensus list of m6A sites in HOTAIR-expressing breast cancer cell lines.** X indicates an m6A site detected in 2+ replicates in the cell line noted.
(DOCX)

**S3 Table. HOTAIR m6A sites identified in breast cancer cell lines generated from patient-derived xenografts.** Each meCLIP experiment is listed with the UCD cell line, detection of m6A783, number of raw reads over HOTAIR A783, and number of C➔T conversions following A783. The last column lists the transcripts per million and raw reads obtained in RNA seq experiments in [33].
(DOCX)

**S4 Table. List of shRNAs and ORFs used in this study.**
(DOCX)

**S5 Table. Oligonucleotides used for constructing pBABE-Puro-A783U_HOTAIR, pBABE-Puro-A782U_HOTAIR, pBABE-Puro-A783C_HOTAIR, pBABE-Puro-C784U_HOTAIR, and dCasRX-YTHDC1.**
(DOCX)

**S6 Table. Oligonucleotides used for m6A eCLIP.**
(DOCX)

**S7 Table. Oligonucleotides used for qPCR.**
(DOCX)

**S8 Table. Plasmids and oligonucleotides used for in vitro m6A methylation experiments.**
(DOCX)

**S1 Raw Images. Uncropped versions of blots in Fig panels 2F, 3A, 4D, 4G, 6B, S2C, S6A, and S6B.**
(PDF)

## Acknowledgments

We thank lab members April Griffin, Sydney Vik, and Gabriela Padilla for technical support. We thank Maggie M. Balas for figure advice and preparation, Carol Sartorius and Jessica Finlay-Schultz for providing recent patient-derived breast cancer cell lines, Rafael Margueron for providing MS2-tethered HOTAIR cell lines used in this study, Chuan He for providing METTL3 and METTL14 expression plasmids, Patrick Hsu for providing the dCasRX plasmids, and John Rinn, Suja Jagannathan, Neelanjan Mukherjee, and Maria Aristizabal for their suggestions on the manuscript.

## Author Contributions

**Conceptualization:** Allison M. Porman, Jennifer K. Richer, Aaron M. Johnson.

**Data curation:** Allison M. Porman, Aaron M. Johnson.

**Formal analysis:** Allison M. Porman, Justin T. Roberts, Emily D. Duncan, Michelle M. Williams, Jennifer K. Richer, Aaron M. Johnson.

**Funding acquisition:** Allison M. Porman, Aaron M. Johnson.

**Investigation:** Allison M. Porman, Justin T. Roberts, Emily D. Duncan, Madeline L. Chrupcala, Ariel A. Levine, Michelle A. Kennedy, Aaron M. Johnson.

**Methodology:** Allison M. Porman, Justin T. Roberts, Emily D. Duncan.

**Project administration:** Allison M. Porman, Aaron M. Johnson.

**Software:** Justin T. Roberts.

**Supervision:** Allison M. Porman, Aaron M. Johnson.

**Validation:** Allison M. Porman.

**Visualization:** Allison M. Porman, Justin T. Roberts, Emily D. Duncan.

**Writing – original draft:** Allison M. Porman, Aaron M. Johnson.

**Writing – review & editing:** Allison M. Porman, Justin T. Roberts, Emily D. Duncan, Madeline L. Chrupcala, Ariel A. Levine, Michelle A. Kennedy, Michelle M. Williams, Jennifer K. Richer, Aaron M. Johnson.

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
