## [Editor Report · Decision Letter 0]

27 Aug 2021

Dear Dr Johnson, 

Thank you for submitting your manuscript entitled "A single N6-methyladenosine site in lncRNA HOTAIR regulates its function in breast cancer cells" for consideration as a Research Article by PLOS Biology.

Your manuscript has now been evaluated by the PLOS Biology editorial staff as well as by an academic editor with relevant expertise and I am writing to let you know that we would like to send your submission out for external peer review.

Please re-submit your manuscript within two working days, i.e. by Aug 29 2021 11:59PM.

Kind regards,

Richard

Richard Hodge, PhD

Associate Editor, PLOS Biology

rhodge@plos.org

PLOS

---

## [Decision Letter · Decision Letter 1]

22 Sep 2021

Dear Dr Johnson,

Thank you for submitting your manuscript "A single N6-methyladenosine site in lncRNA HOTAIR regulates its function in breast cancer cells" for consideration as a Research Article at PLOS Biology. Your manuscript has been evaluated by the PLOS Biology editors, an Academic Editor with relevant expertise, and by three independent reviewers.

The reviews are attached below. You will see that the reviewers find your conclusions interesting and the data well executed, but also raise overlapping concerns with the overall strength of the data implicating a role for YTHDC1 in the model. In addition, Reviewer #1 raises concerns with the physiological relevance of the model system used and asks that key pieces of data are replicated in more clinically relevant samples. Reviewer #3 also asks for two additional control experiments to strengthen the claims that the specific A783 methylated site is critical for the model presented.

In light of the reviews, we will not be able to accept the current version of the manuscript, but we would welcome re-submission of a much-revised version that takes into account the reviewers' comments. We cannot make any decision about publication until we have seen the revised manuscript and your response to the reviewers' comments. Your revised manuscript is also likely to be sent for further evaluation by the reviewers.

We expect to receive your revised manuscript within 3 months. 

**IMPORTANT - SUBMITTING YOUR REVISION**

*Re-submission Checklist*

*Published Peer Review*

*PLOS Data Policy*

*Blot and Gel Data Policy*

Sincerely,

Richard

Richard Hodge, PhD

Associate Editor, PLOS Biology

rhodge@plos.org

PLOS

REVIEWS:

Reviewer #1: In this manuscript, Porman et al investigate the role of m6A in HOTAIR-mediated transcriptional repression. The authors report that methylation primarily of a single residue results in chromatin association in a manner mediated by YTHDC1, which culminates in HOTAIR stabilization, transcriptional repression, increased proliferation and potentially reduced survival in cancer patients. 

There are multiple dimensions of this manuscript that I enjoyed. This work is among the few in literature where a rigorous attempt is made to connect m6A at a single position to both a molecular and a physiological outcome. Also the depth of mechanistic examination performed by the authors is impressive. The manuscript is also well and concisely written, and I enjoyed the fact that at various instances where observations were inconsistent with their model, the authors explicitly pointed this out and discussed it.

Nonetheless, there remain points of lingering concern, where both clarifications and additional experimentation could help substantiate and consolidate the results. 

Major points:

Our first concern pertains to the presence of m6A at the position reported by the authors. As also noted by the authors, the position that they detect is an atypical one - it lacks a classical 'DRACH' consensus motif, and as such would be predicted to not undergo modifications at all, or at low stoichiometries. This position was exclusively (yet consistently) identified on the basis of eCLIP experiments on WT samples. Such eCLIP experiments were not performed following knockdown of methyltransferase machinery components. Moreover, the authors report no change in HOTAIR immunoprecipitation efficiency using an anti-m6A antibody in A783U mutants lacking m6A at this position (we note that the eCLIP signature is lost in this mutant). Given the centrality of the claims concerning m6A at this position, in our view it is critical that this point be confirmed both on the basis of additional controls (e.g. eliminating methyltransferase components) and ideally also approaches.

An additional major concern is that a large body of the conclusion rests on a single mutant (A783U). While the tacit assumption is that this particular mutant differs only in methylation status, it cannot be ruled out - in particular for a structured molecular such as HOTAIR - that this particular mutations impacts the structure or folding of HOTAIR, which underlie some (or all) observations. The conclusions would be hugely boosted by introducing two additional mutants, perturbing the 'C' immediately downstream of the methylated site and the 'A' upstream, and demonstrating that these mutants phenocopy the A783U mutants in growth and chromatin localization.

More minor points:

An aspect I found somewhat confusing in this manuscript, and which also renders the interpretation of many of the experiment confusing, is the causal relationship between YTHDC1 and HOTAIR expression levels. In Figure 1D the authors demonstrate that mutant HOTAIR gives rise to reduced levels of YTHDC1 than WT HOTAIR, suggesting that YTHDC1 is stabilized by HOTAIR. Yet, the model argued by the authors - and for which they also display evidence - is that YTHDC1 is stabilized by HOTAIR. So is there a circular relationship here? This should be clarified.

Related to this aspect, we found it confusing that if these genes are positively correlated with another, and increased levels of both genes are required for increased proliferation (Figure 2E), then how does this connect to the Kaplan Meier curves, according to which the genes appear to display an opposite effect on survival (for YTHDC1, reduced levels are associated with poorer survival, for HOTAIR with improved survival)? More generally, we did not feel that this section (Figure 3) was very compelling. In general the associations with gene expression levels were weak, and the differences in association with HOTAIR between high YTHDC1 and low YTHDC1 were, again, weak. We do not think these are of critical importance for the manuscript, and leave this to the author's discretion. 

I loved the experiments in Figure 4E-F dissecting chromatin localization of WT and mutant HOTAIR following perturbation of YTHDC1 levels, and the results for the mutant seem very compelling. However, I find it difficult to accept the explanation by the authors for the complete lack of effect of YTH depletion on WT HOTAIR. A key component of their model is that YTHDC1 binding to A783 of HOTAIR gives rise to chromatin localization. Yet, in this key experiment, depletion of YTHDC1 gives rise to even slightly increased chromatin localization. The authors suggest this could have to do with insufficient knockdown, yet clearly the levels of knockdown are sufficient to destabilize HOTAIR and to give rise to phenotypes. Could this possibly be a normalization issue, potentially caused by the reduced levels of HOTAIR following the knockdown? Assuming it is the knockdown efficiencies, can the authors perturb YTHDC1 using other means, and confirm that under this scenario chromatin localization is reduced?

In Figure 5 the authors seek to draw a direct connection between HOTAIR and YTHDC1-mediated transcriptional repression using a reporter construct. Yet, in these experiments the authors omitted the control of the A783U mutant. While this result therefore establishes some requirement of YTHDC1 in mediating repression, it fails to associate this with HOTAIR methylation. 

Finally, the tethering experiments conducted in Figure 7 provide substantial support for the model proposed by the authors, demonstrating that chromatin association of a HOTAIR mutant can be rescued by tethering it to YTHDC1. It is unclear to me, though, why in these experiments no stabilization of HOTAIR is observed. In addition, can the proliferation phenotype - that is lost in the the A783U mutant - also be rescued in these experiments, which would elegantly combine the molecular and physiological levels?

Reviewer #2: The manuscript titled "A single N6-methyladenosine site in lncRNA HOTAIR regulates its function in breast cancer cells" by Porman et al. reports the finding that a particular m6A modification site on HOTAIR is important for proliferation and invasion of triple-negative breast cancer cells. The authors present data to suggest a mechanistic model where YTHDC1 is the reader that is recruited to the particular methylation site (A783) and that it impacts chromatin association and expression of HOTAIR. Overall, the big-picture model that a single m6A site on HOTAIR can impact gene expression is supported well and interesting. Here are some comments and suggestions for revising the manuscript. 

1. While the importance of A783 is clear, the evidence for YTHDC1 to be singularly playing a role in HOTAIR-mediated breast cancer progression is not as strong. YTHDC1 binds most m6A-modified RNAs, and RIP procedure might result in associations that do not occur in the cell (eg. due to different localization). Is there evidence that YTHDC1 binds A783 in cells? 

2. YTHDC1 seems to have a relationship with HOTAIR, given the expression level changes. Do the authors see no such relationship for other readers in the YTH-family?

3. The in vitro experiments to show that YTHDC1 binds A783 of HOTAIR also have some issues. How did the authors confirm that METTL3/14 methylate HOTAIR at the correct position (A783) in vitro? 

4. For the m6A and YTHDC1 RIP experiments, the authors refer to the other m6A sites in HOTAIR that are modified in MDA-MB-231 cells for the lack of difference between WT and mutant. How about MCF7 cells where the authors said that they only found a single site?

5. Tethering YTHDC1 to A783U HOTAIR is a neat experiment. How does it affect cell growth and invasion when the need for m6A is bypassed this way? If the authors can see direct effects, that would strengthen the presented model. 

6. In multiple figures, the legend does not indicate what kind of values are being reported. Please state whether it is a mean or median, from how many replicates. Also, p values should always be indicated with what kind of statistical test.

7. How do the authors reconcile the difference between Fig 2E and 3F? Is YTHDC1 level important or not for WT HOTAIR?

8. In Fig 4B, how did the authors ensure that the amount of bait RNA is similar for all samples? 

9. Can the authors distinguish between direct and indirect targets of HOTAIR for transcriptional regulation? 

Reviewer #3: The authors study m6A on the lncRNA HOTAIR in the context of breast cancer. They performed an enormous amount of work using a range of techniques and obtain some interesting findings. However, several questions withing the study remain unresolved and the proposed mechanism is somewhat convoluted. 

1. If HOTAIR is highly expressed in TNBC patients, why do MDA-MB-231 (or MCF-7) cells not have high levels? While the authors are trying to make a case for their choice of model system, using ectopic expression of HOTAIR in 231 cells, I remain skeptical that this is indeed reflective of the in vivo situation in patients, in particular as m6A status in patients has not been studied here. A breast cancer cell line (and/or primary cells) with endogenously high HOTAIR levels may be a better model system. Replication of some key data in more clinically relevant cells or patient tissue could substantially elevate the findings.

2. Figure 1D does not have a "yellow to red scale" as indicated in the legend, it seems like the color scheme has been changed without updating the legend

3. Why does endogenous HOTAIR only have one m6A site in MCF7 while overexpressed HOTAIR has 8 m6A sites? Is this dependent on expression level or cell line or BC subtype?

4. Phenotypic differences between cells transfected with HOTAIR and the A783U mutant seem rather modest overall, and there is no evidence that this is reflective of a patient tumour. 

5. The authors switch back and forth between the two cell lines, while unclear if findings in one also hold up in the other. Is there a phenotypic effect on cell growth in MCF-7 as well? Does hnRNP B1 bind HOTAIR m6A783 in MDA-MB-231 cells? Does YTHDC1 oe/kd impact on growth in MCF7 cells?

6. How do WT / control transfected MDA-MB-231 behave when YTHDC1 is overexpressed or knocked down? HOTAIR won't be the only target, and the effects may be indirect on target genes of HOTAIR. 

7. Correlation between YTHDC1 and HOTAIR in patients may be "very modest" due to the fact that the two genes simply don't correlate. The survival curves are all indirect measures, and the patient data is overall somewhat hard to interpret in the context of this story as patients' m6A status is unknown.

8. Since binding of YTHDC1 does not change whether the A783U mutant is methylated or not, it is questionable whether a) this residue is really as important as the rest of the manuscript suggests and b) whether interactions with YTHDC1 are mediated by A783me at all. A bit confusing in light of the difference in cell doubling data and the difference in chromatin association. The given explanation regarding highly constitutive interactions and binding sites of different affinities that change in the mutant seem all rather speculative. What happens to breast cancer cells when the other m6A residues are mutated?

---

## [Decision Letter · Decision Letter 2]

26 Apr 2022

Dear Dr Johnson,

Thank you very much for submitting a revised version of your manuscript "A single N6-methyladenosine site in lncRNA HOTAIR regulates its function in breast cancer cells" for consideration as a Research Article at PLOS Biology. Please accept my apologies for the delay in getting back to you with our decision. This revised version of your manuscript has been evaluated by the PLOS Biology editors, the Academic Editor and the original reviewers.

The reviews are attached below. As you can see, whilst Reviewer #3 is now satisfied with the revision, Reviewers #1 and #2 still raise overlapping concerns that the effect of the A783U point mutation may not be dependent on m6A methylation. Specifically, Reviewer #1 notes that an orthogonal validation using an additional mutant should be performed to demonstrate site-specificity and Reviewer #2 raises concerns with the strength of the RIP data showing the interaction of YTHDC1 with HOTAIR-A783.

After discussing the reviews at length with the Academic Editor, we feel that these experiments would improve the manuscript and are important to provide direct support for the claims that the single methylation event regulates the observed phenotypes. However, we will not make the second concern of Reviewer #1 regarding the folding of HOTAIR an editorial requirement. This can be addressed by caveating and discussing the alternative interpretations in the manuscript text.

In light of the reviews, we will not be able to accept the current version of the manuscript, but we would welcome re-submission of a revised version that takes into account the reviewers' comments. We cannot make any decision about publication until we have seen the revised manuscript and your response to the reviewers' comments.

We expect to receive your revised manuscript within 3 months. Please email us (plosbiology@plos.org) if you have any questions or concerns, or would like to request an extension. At this stage, your manuscript remains formally under active consideration at our journal; please notify us by email if you do not intend to submit a revision so that we may end consideration of the manuscript at PLOS Biology.

**IMPORTANT - SUBMITTING YOUR REVISION**

*Re-submission Checklist*

*Published Peer Review*

*PLOS Data Policy*

*Blot and Gel Data Policy*

Sincerely,

Richard

Richard Hodge, PhD

Associate Editor, PLOS Biology

rhodge@plos.org

REVIEWS:

Reviewer #1: In our previous review we had two key comments. First, the lack of a classical m6A consensus motif at the identified site, combined with lack of reduced signal in m6A-IPs following its point-mutation, raised concerns as to whether the site was actually modified, and we suggested the use of orthogonal measurements. Second, we suggested an additional mutant that would disrupt methylation, to rule out structural or indirect effects of the single mutant explored in the manuscript.

The authors have performed neither of these suggestions. With respect to the first point, the authors now report that they find reduced m6A signal following depletion of METTL3 or WTAP. This, however, does not address the concern. My concern was not that HOTAIR is not methylated, nor that it is not methylated by the METTL3 complex. My concern was that it is methylated at a site distinct from the one reported by the authors, and that the effect of their point-mutant may not be dependent on m6A. The authors argue that these experiments were done in cells where this site was the only one that had been detected. Yet, miCLIP studies have notoriously low sensitivity. In a typical experiment few thousand methylated residues are pinpointed, out of tens of thousands, and hence while one can generally rely on their specificity, the same cannot be assumed for their sensitivity.

The authors also argue that a second mutant is not necessary, given that the methylated site is in a structurally open region. I cannot follow this argumentation. Such an open structure may be required for HOTAIR function. Potentially the point-mutant gives rise to a more closed structure, and in this manner may impact HOTAIR. 

Thus, I remain with some lingering concerns regarding the validity of the identified sites, and hence also regarding interpretation of the results in the mutants. Having said this, already in our first review we were relatively enthusiastic about this paper. We also do appreciate that another mutant - and follow up experiments - would constitute quite a bit of additional work. We thus leave the necessity of these additional controls to the discretion of the editor.

Reviewer #2: The manuscript is improved. However, there are still some questions. 

1. The figure legends have improved and contain information about replicates. However, it would help the readers to judge the strength of the data if it was easier to estimate the SEM or SDM, through error bars. I suggest that instead of the box and whisker plots, the authors provide bar graphs with error bars when possible. Many of the graphs do not have the statistical tests, and just having the error bars would help. 

2. Although the authors have revised this section, I am still not convinced that the specific event of YTHDC1 interaction with m6A783 is what leads to the dramatic changes. For RIP-qRT-PCR, the data in 2D is helpful but also seems not thorough. Do the authors detect multiple methylation sites when they perform a similar experiment with MDA-MB-231 cells? What about with the mutant A783U HOTAIR? When the authors check other regions of HOTAIR, are they also all free of m6A in MCF7 cells? These seem to be readily feasible experiments that could support the claims. 

3. Do the authors know what % of HOTAIR transcripts have m6A783? Or what % of HOTAIR is bound to YTHDC1 vs not?

4. Line 213, "the only difference in methylation … should be at A783". This is an assumption, and might be better to be reworded. It would also help to include a comment that such an in vitro methylation experiment can generate many more m6A sites than in cells. Figure S2D and S2E are missing. In S3?

Reviewer #3: Porman et al have significantly strengthened their manuscript with additional experiments and explanations. The authors are showing an interesting new molecular mechanism in this manuscript, which expands upon the current knowledge in the field. My main point of weakness for this paper remains the absence of clear evidence that this mechanism is in fact relevant in the clinic, however, this does not deter from my overall excitement for the study.

In particular, I was pleased to see that the binding of HOTAIR and YTHDC1 was explored in more detail, and that analysis of the 6xAU and 14xAU mutations was added. Both experiments further strengthen the proposed mechanism.

While it is reassuring that the authors were able to identify m6A783 in 1/3 cell line derived from actual breast tumors, the clinical significance (if any) remains unclear. As mentioned in my previous comments, the cell lines used here plus additional ones tested all seem to express HOTAIR at levels too low to quantify/detect m6A, which seems somewhat conflicting with the overarching hypothesis that HOTAIR is highly expressed in TNBC. 

I appreciate that the authors have dialed down the potential clinical meaning of m6A783 for TNBC and removed some of the clinical data on YTHDC1.

The study now focuses more on its strengths, acknowledging that is a mechanistic investigation in cell lines, with potential to be relevant in patients as shown in one patient-derived cell line. It may be too early days to state whether the described phenomenon may or may not have any meaning in the clinic based on the data presented. I would certainly not go so far as to suggest therapeutic potential without substantiating the findings in clinically relevant models / datasets first - perhaps the last sentence in the abstract can be rephrased more carefully. 

Overall, the manuscript has improved, and I would endorse publication.

---

## [Editor Report · Decision Letter 3]

13 Oct 2022

Dear Aaron,

Thank you for your patience while we considered your revised manuscript "A single N6-methyladenosine site in lncRNA HOTAIR regulates its function in breast cancer cells" for publication as a Research Article at PLOS Biology. This revised version of your manuscript has been evaluated by the PLOS Biology editors and the Academic Editor.

Based on our Academic Editor's assessment of your revision, I am pleased to say that we are likely to accept this manuscript for publication, provided you satisfactorily address the following data and other policy-related requests that I have provided below (A-G):

(A) We would like to suggest the following modification to the title:

‘A single N6-methyladenosine site regulates lncRNA HOTAIR function in breast cancer cells’

(B) You may be aware of the PLOS Data Policy, which requires that all data be made available without restriction: http://journals.plos.org/plosbiology/s/data-availability. For more information, please also see this editorial: http://dx.doi.org/10.1371/journal.pbio.1001797

- Supplementary files (e.g., excel). Please ensure that all data files are uploaded as 'Supporting Information' and are invariably referred to (in the manuscript, figure legends, and the Description field when uploading your files) using the following format verbatim: S1 Data, S2 Data, etc. Multiple panels of a single or even several figures can be included as multiple sheets in one excel file that is saved using exactly the following convention: S1_Data.xlsx (using an underscore).

- Deposition in a publicly available repository. Please also provide the accession code or a reviewer link so that we may view your data before publication.

Figure 1B-D, 1F-H, 2A-D, 2G, 3B-G, 4B-C, 4E-F, 4H-I, 5B-C, 5E, 6C-G, S2A-E, S3B-D, S4D-F, S5A-D, S6D, S7A-B, S8D

(C) Please also ensure that each of the relevant figure legends in your manuscript include information on *WHERE THE UNDERLYING DATA CAN BE FOUND*, and ensure your supplemental data file/s has a legend.

(D) Please note that per journal policy, the model system/species studied should be clearly stated in the abstract of your manuscript. 

We expect to receive your revised manuscript within two weeks.

*Published Peer Review History*

*Press*

Best wishes,

Richard

Richard Hodge, PhD

Associate Editor, PLOS Biology

rhodge@plos.org

PLOS

---

## [Editor Report · Decision Letter 4]

24 Oct 2022

Dear Aaron,

On behalf of my colleagues and the Academic Editor, Jeff Coller, I am pleased to say that we can accept your manuscript for publication, provided you address any remaining formatting and reporting issues. These will be detailed in an email you should receive within 2-3 business days from our colleagues in the journal operations team; no action is required from you until then. Please note that we will not be able to formally accept your manuscript and schedule it for publication until you have completed any requested changes.

PRESS

Best wishes, 

Richard

Richard Hodge, PhD

Associate Editor, PLOS Biology

rhodge@plos.org

PLOS
